computational mechanics/biomimetics/biophysics

butterfly flight, wing-pitch motion, induced flow, wake-capture effect, forward propulsion

**Author for correspondence:**
Jing-Tang Yang
e-mail: jtyang@ntu.edu.tw

# Beneficial wake-capture effect for forward propulsion with a restrained wing-pitch motion of a butterfly

You-Jun Lin, Sheng-Kai Chang, Yu-Hsiang Lai and Jing-Tang Yang

Department of Mechanical Engineering, National Taiwan University, Taipei 10617, Taiwan

Y-JL, 0000-0003-0985-6027; S-KC, 0000-0003-3967-5374; Y-HL, 0000-0003-2953-2812

Unlike other insects, a butterfly uses a small amplitude of the wing-pitch motion for flight. From an analysis of the dynamics of real flying butterflies, we show that the restrained amplitude of the wing-pitch motion enhances the wake-capture effect so as to enhance forward propulsion. A numerical simulation refined with experimental data shows that, for a small amplitude of the wing-pitch motion, the shed vortex generated in the downstroke induces air in the wake region to flow towards the wings. This condition enables a butterfly to capture an induced flow and to acquire an additional forward propulsion, which accounts for more than 47% of the thrust generation. When the amplitude of the wing-pitch motion exceeds 45°, the flow induced by the shed vortex drifts away from the wings; it attenuates the wake-capture effect and causes the butterfly to lose a part of its forward propulsion. Our results provide one essential aerodynamic feature for a butterfly to adopt a small amplitude of the wing-pitch motion to enhance the wake-capture effect and forward propulsion. This work clarifies the variation of the flow field correlated with the wing-pitch motion, which is useful in the design of wing kinematics of a micro-aerial vehicle.

## 1. Introduction

The typically large frequency of wing beating by insects might create difficulty for a mechanical structure to withstand the vibration caused by rapid flapping and decrease the durability of a micro-aerial vehicle (MAV). A butterfly with a small wing-beat frequency thus becomes an effective biomimetic model for the design of an MAV. This frequency of a butterfly is only about 10 Hz, much less than most insects such as a bumblebee, 152 Hz [1], or a hawk moth, 26.1 Hz [2], or a cicada, 46.5 Hz [3]. Besides

gliding flight [4], a butterfly has a notable oscillatory motion of the body [5,6] that is unique among insects. The fore and hind wings of a butterfly partially overlap, so resemble flapping with one pair of broad wings [7]. When a butterfly flaps its wings, the instantaneous pressure at the tip of the fore wing can attain nearly 10 Pa, or 10 times that of the wing loading [8], which enables a butterfly to alter its direction of flight in only a few flapping periods. The great manoeuvrability and small flapping frequency of a butterfly might provide abundant fertile material for scientists to design an MAV.

Insects experience rapid wing supination and pronation at a stroke reversal, which is named as a wing-pitch motion and is an important unconventional transient mechanism to generate an aerodynamic force [9]. Many researchers proved that the wing-pitch motion is a critical mechanism among flying insects [10–12]. The amplitude of the wing-pitch angle of a butterfly, which is about 50° [13], is however, much smaller than that of other insects, such as a cicada, 100° [14], a dragonfly, 135° [15], a hawkmoth, 95° [16], a mosquito, 145° [17] or a fruit fly, 135° [18]. Dudley [19] assumed that the wing-pitch motion of a butterfly is not obvious; Fei & Yang [5] likewise maintained that the wing-pitch angle is small; Tanaka & Shimoyama [20] neglected the wing-pitch motion in its butterfly-like flapping mechanism; Senda et al. [21] declared that a butterfly can be considered to implement an almost one degree-of-freedom flapping mechanism. The above literature shows that many researchers neglected the wing-pitch motion of a butterfly, but there are still articles that take this motion into account [6,13,22,23]. Bode-Oke & Dong [13] stated that the upstroke of a butterfly is characterized by not only body oscillation but also wing-pitch motion to aid the wings. As the rapid supination and pronation is an important mechanism for insects [12], we considered that the restrained wing-pitch motion adopted by a butterfly might exist as an additional mechanism that has not been discovered. Why a butterfly flies with a small amplitude of its wing-pitch motion is worthy of exploration.

During the wing-pitch motion of an insect, there are mainly two unsteady effects—rotational circulation and wake capture. For rotational circulation, an additional circulation is reported to be generated around the wings [10]; this circulation causes a rotational force that either increases or decreases the net force according to the timing of the wing-pitch motion relative to each wing stroke [11]. Dickinson et al. [12] proposed a self-made flapping mechanism and indicated that an advanced wing-pitch motion enhances the production of lift with the rotational force. The rotational force has a positive correlation with the wing-pitch angular velocity (called a pitch rate) in a quasi-steady model [24], and is also proportional to the circulation in the surrounding air [9]. The wake capture occurs at the beginning of each half-stroke and generates either a beneficial or an adverse effect depending on the situation [25]. As the wings flap into a wake, the flow induced by a shed vortex towards the wings increases the relative velocity and lift on the wings [11,26]. By contrast, when the vortex of the preceding stroke generates a downwash flow, the wings flapping into the wake cause a decreased aerodynamic force [27,28]. Also, as a wing encounters a shed leading-edge vortex (LEV) that slips along the wing surface, it creates a vortex suction on the windward side of the wing and decreases the lift [29,30]. Srygley & Thomas [31] reported that a flow field varies during the flying sequence of a butterfly; they detected a subtle change in the wing kinematics in that that butterfly might choose to apply, or not to apply, the wake-capture effect. The wing kinematics were neither mentioned nor combined with the flow field in that study. We suspect that the variation of a wake-capture effect arises because of the varied wing-pitch motion. The effect of the wake capture is a function that combines with the wing kinematics and the flow structure around the wings [12,26,32]. For insects that have a small Reynolds number, the unsteady flow field around the flapping wing is even more sensitive to the kinematic parameters [33]. If we can unveil a relation between an unsteady flow field and the kinematics, this knowledge would increase our understanding of the flight of a butterfly and improve the wing-kinematic design of an MAV.

In this work, we measured the wing kinematics from a real flying butterfly, and created a numerical model to investigate the wing-pitch motion in relation to the flow field. The amplitude of the wing-pitch motion is parametrized in the model. The flow field, the vortex structure and the differences in force generation among varied wing-pitch motions are discussed. We clarify the effect of the varied wing-pitch motion on the flight of a butterfly, and provide one essential aerodynamic feature for the fact that a butterfly adopts a small amplitude of the wing-pitch motion for flight.

# 2. Material and methods

## 2.1. Experiment set-up and definition of the coordinate system

To measure a real, freely flying butterfly (Tirumala septentrionis), we placed a specimen inside a transparent experimental chamber (Plexiglas), and deployed two synchronized high-speed cameras

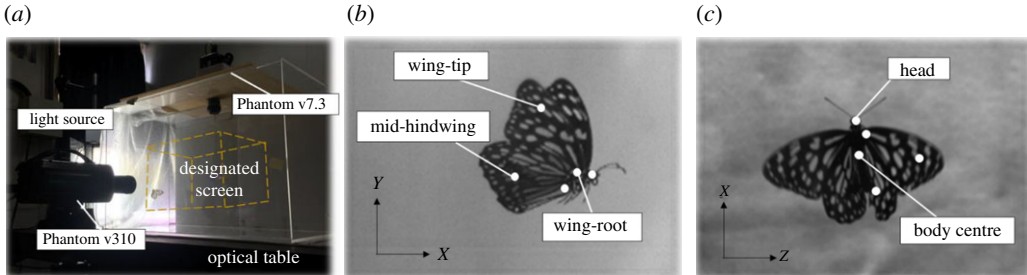

**Figure 1.** (*a*) Experimental environment. (*b*) Five characteristic points—wing-root, wing-tip, mid-hindwing, (*c*) body centre and head—were selected and tracked to implement the post-analysis.

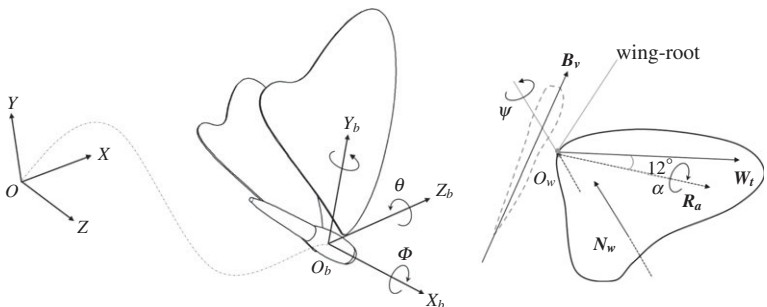

**Figure 2.** Description of coordinate systems for analysis of the motion of a butterfly. Point $O$ is the reference to the laboratory; $O_b$ on behalf of the origin of the coordinate system is fixed on the centre of the body mass; $O_w$ refers to the origin of the coordinate system fixed at the point of the wing-root.

(Phantom v7.3 and Phantom v310, operated at rate 1000 frame s$^{-1}$ with resolution 1024 × 1024 pixels) with orthogonal viewing directions to record the flying motion of that butterfly (figure 1*a*). A lamp was set to attract a butterfly to fly forward. Four butterfly samples obtained from Mu Sheng Insect Museum in Nantou, Taiwan, were used. A butterfly was placed at the entrance of the experimental chamber and let fly freely. The high-speed cameras captured the duration of the flight as the butterfly flew in the designated screen (33 × 25 cm in side view and 40 × 30 cm in top view). In general, three or four wing-beat cycles were captured in each film. A complete and clear wing beat was chosen for analysis; the frequency was obtained on calculating the number of frames included. To refine the recorded video for forward flight with no turning, we discarded those videos in which a deviation more than 10° to the left or right occurred and retained 16 films in which the flight was directly forward. The speckles on the wings of the butterflies served as characteristic points to derive the kinematic motion. As the fore and hind wings partially overlap, referring to published research [7,19], we simplified the wings as one pair of broad wings. Three points—wing-root, wing-tip and mid-hindwing—portray the wings; two points depict the body (figure 1*b*). As the maximum deformation of the wings occurs at their edges, the characteristic points of wing-tip and mid-hindwing were chosen inside the wing surface rather than at the outer edge to mitigate the influence of wing flexibility on the measurement of wing-pitch angle. We first specified these characteristic points in the sequences of captured images and derived the positions of these points; a program was written (in MATLAB code) to calculate the angles (§2.2).

To describe the motion of a flying butterfly, we adopted a wing coordinate system ($X_wY_wZ_w$), a body-fixed coordinate system ($X_bY_bZ_b$) and a global coordinate system ($XYZ$) (figure 2); a butterfly is allowed to fly freely as loaded. The origin of the wing coordinate system ($O_w$) is set at the base of the wings; the three axes of the coordinate system are specified as the direction of a chord line ($X_w$), the vector normal to the wing surface ($Y_w$) and the span of a wing ($Z_w$). The body-fixed coordinate system is a non-inertial system; its origin ($O_b$) coincides continuously with the centre of mass of the butterfly. The three axes of the body-fixed system are represented as pitch ($Z_b$), roll ($X_b$) and yaw ($Y_b$) of the body. The origin of the global coordinate system ($O$) is set as the centre of mass of a butterfly at the moment of initiation of flight. The butterfly moves forward along axis $X$; negative axis $Y$ is the direction of the gravity force; axis $Z$ is the reference axis for the rotation of the body. The body angle is denoted $\theta$, which is specified according to the centre line of the thorax and plane $XZ$; the body is parallel to plane $XZ$ when $\theta$ is zero, perpendicular when $\theta$ is 90°.

**Table 1.** Biological information of a butterfly (*T. septentrionis*) in simulation based on experimental data.

|  | simulation | experiment (average ± s.e.m.) |
|---|---|---|
| total mass (g) | 0.35 | 0.35 ± 0.03 |
| wing-beat frequency (Hz) | 11.00 | 11.03 ± 0.28 |
| wing length (mm) | 46.04 | 45.25 ± 0.90 |
| mean wing chord (mm) | 20.11 | 19.97 ± 0.19 |
| body length (mm) | 25.00 | 25.70 ± 1.13 |
| aspect ratio | 2.29 | 2.27 ± 0.03 |

## 2.2. Definition of wing angles

To quantify the flapping, sweeping and wing-pitch angles of the wings from the experiment, we applied the Euler angles (flap–sweep–pitch) in our work. In figure 2, $R_a$ represents the wing-pitch axis that has angle 12° with wing-tip vector $W_t$ (see the detail in electronic supplementary material, figure S2); $W_t$ is the vector between the wing-tip and wing-root points. The flapping angle $\phi$ is defined as an angle between $R_a$ and vector $Z \times B_V$

$$\phi(t) = \cos^{-1} \frac{R_a(t) \cdot (Z \times B_v(t))}{|R_a(t)||(Z \times B_v(t))|}, \tag{2.1}$$

in which $B_v$ represents the body vector. The vector is related to $t$, except for axis $Z$, which is invariably equal to $Z = (0, 0, 1)$. The sweeping angle $\psi$ is the complementary angle between $W_t$ and $B_v$

$$\psi(t) = \frac{\pi}{2} - \cos^{-1} \frac{W_t(t) \cdot B_v(t)}{|W_t(t)||B_v(t)|}, \tag{2.2}$$

in which $W_t$ is the wing-tip vector. The wing-pitch angle $\alpha$ is calculated as the angle between two vectors, of which one is normal to the wing plane $N_w$; the other is normal to the plane before the wing rotates, which is vector $R_a \times B_v$

$$\alpha(t) = \cos^{-1} \frac{N_w(t) \cdot (R_a(t) \times B_v(t))}{|N_w(t)||(R_a(t) \times B_v(t))|}. \tag{2.3}$$

## 2.3. Numerical model and simulation scheme

We applied a numerical method to address the flying behaviour. Two workstations (Intel Xeon W-3175X 3.10 GHz, RAM–128 GB; AMD Ryzen R9-3990X 2.90 GHz, RAM–128 GB) were used for numerical solving. The wing model followed the real size of a blue-tiger butterfly (*T. septentrionis*) with software (Solidworks). The thorax and abdomen were combined to become a body. To analyse the effect of the wing-pitch motion among varied cases (§4), we assumed the wings to be flat plates to separate the effect of the wing-pitch motion from the wing flexibility. Many authors presented satisfactory results using flat plates [34–38]; the use of a flat plate does not affect the natural trend of flight of a butterfly [5,39].

Table 1 shows a comparison of the geometry and features between the experimental butterflies and the butterfly model. The wing model had wing length (*S*) 46.04 mm, mean wing chord ($\bar{c}$) 20.11 mm and thickness 2.5% $\bar{c}$; the radius of gyration ($r_2$) was 24.51 mm, which is 53% of the wing length.

A numerical method was applied with commercial software (ANSYS FLUENT); the semi-implicit method for pressure-linked equations-consistent (SIMPLEC) was chosen to solve the pressure and velocity fields; the Green–Gauss node-based method was applied for the spatial discretization. As the Reynolds number of the flying butterfly is about $10^3$–$10^4$, we assumed the flow to be incompressible and laminar. The air is a Newtonian fluid of density $\rho_f = 1.23$ (kg m$^{-3}$) and viscosity $\mu = 1.79 \times 10^{-5}$ (Pa s). Our model was solved in a relative coordinate system in which the origin coincided continuously with the centre of mass of the butterfly. The relative coordinate moved translationally with the centre of mass of a butterfly, and did not rotate with the oscillatory body of a flying butterfly. This condition makes the model experience an acceleration $a_v$ caused by a virtual force due

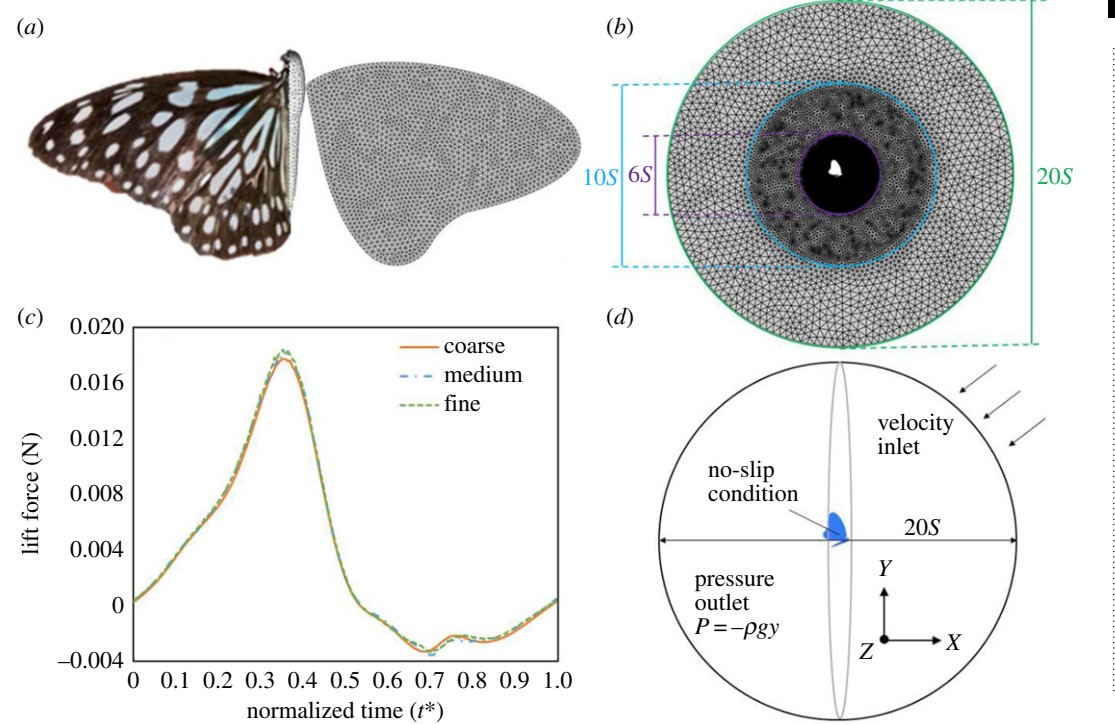

**Figure 3.** (a) Geometry of a butterfly (*T. septentrionis*) and a butterfly model with face sizing. (b) Domain of the fluid calculation with grid points. The space of diameter 6*S* had the greatest density of mesh number. (c) Grid convergence test between coarse grid (5.4 million), medium grid (11.0 million) and fine grid (16.0 million). (d) Physical model of a butterfly and boundary conditions.

to the coordinate transformation. The governing equations are the three-dimensional continuity equation and the Navier–Stokes equation

$$\nabla \cdot \boldsymbol{u} = 0, \tag{2.4}$$

and

$$\rho_f \left( \frac{\mathrm{D}\boldsymbol{u}}{\mathrm{D}t} \right) = -\nabla P + \mu \nabla^2 \boldsymbol{u} + \rho_f \, \boldsymbol{g} + \rho_f \, \boldsymbol{a}_v, \tag{2.5}$$

in which $t$ denotes time, $\boldsymbol{u}$ velocity vector, $\boldsymbol{g}$ gravity force vector and $P$ pressure field. Each flapping cycle was divided into 250 time steps (0.000363 s of each time interval). The numerical model was placed in a spherical domain and was set to fly in incompressible air at 25°C. We divided the domain into three spherical regions of diameters 6*S*, 10*S* and 20*S*; *S* is one wing semi-span equal to 46 mm (figure 3*a*). The outside boundary was divided into two parts; the part behind the butterfly was the pressure outlet, $P = -\rho gy$; the other part was the velocity inlet. The value of the inlet varied with the flight speed of the butterfly model. The surface of the butterfly model had a no-slip condition (figure 3*d*). To improve the quality of the grids, we increased the density of the grid in the inner region (1 mm mesh grids) and the second-layer region (figure 3*b*). We further undertook face sizing (0.5 mm mesh grids) around the butterfly's body and wings to obtain precise solutions. Three grid numbers in figure 3*c* show that the lift force converged as the grid number increased. The difference in the maximum value between the cases of the fine grid and medium grid was less than 1.1%. The total mesh number used in this work was $1.1 \times 10^7$ with tetrahedral grids. Smoothing and remeshing methods were applied in a dynamic mesh strategy.

The thrust and lift coefficients were normalized according to the mean wing-tip velocity $U_{\text{tip}} = 2\phi_A S f$

$$C_T = \frac{F_x}{(1/2)\rho U_{\text{tip}}^2 S \bar{c}} \tag{2.6}$$

and

$$C_L = \frac{F_y}{(1/2)\rho U_{\text{tip}}^2 S \bar{c}}, \tag{2.7}$$

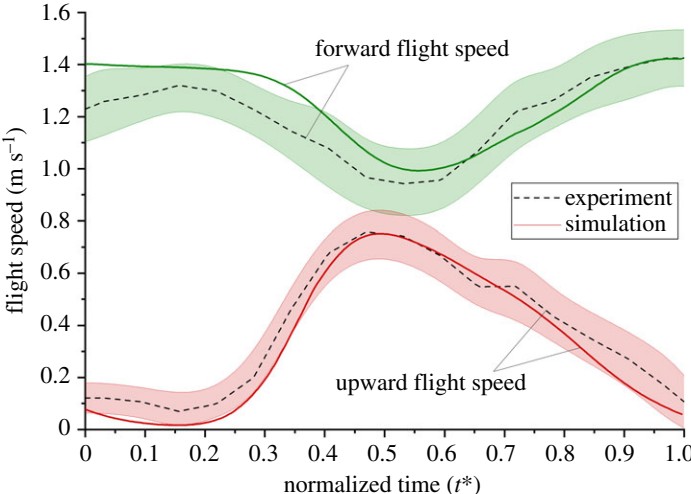

**Figure 4.** Comparison of flight speed between an experimental butterfly and a simulation model in the eighth wing-beat cycle. The shaded areas are standard error of mean (s.e.m.).

in which $\phi_A$ denotes flapping amplitude, $f$ wing-beat frequency; $S$ length of a wing and $\bar{c}$ mean wing-chord length; $F_x$ and $F_y$ are horizontal and vertical aerodynamic forces. The Reynolds number of a butterfly used in this research was based on the wing-tip velocity and the mean chord length, which was 6050 in the experiment and 6130 in the simulation.

## 2.4. Simulation methods for transient flight speed

The simulation method is divided into two parts. For the first part, we simulated a free-flying butterfly, in which the lift and thrust were calculated with the simulator. The kinematic motion (body-pitch and wing motion) in the simulation was given based on the experimental measurement. This method produced the result of §3.5. For the second part, we imported the velocity distribution which was the flight speed of butterfly obtained from the simulation based on real wing and body-pitch kinematic motions. In this way, we make all cases of the butterfly model fly with the same input velocity, and this method produced the results from §§3.2 to 3.4.

### 2.4.1. Free-flight simulation

To reproduce the free-flying trajectory of a butterfly, one must calculate the transient flight speed in the simulation. The simulation method has been published and validated in our previous articles [5,38–42]. In the present work, the initial condition of a butterfly in a simulation is a state in which the flow field is still and has a pressure distribution due to gravity ($P = -\rho g y$); a butterfly is allowed to fly freely as loaded with the experimental motions (equations (3.1)–(3.4)). The mesh updates according to the kinematic motion in each time step. We calculated the aerodynamic forces of the butterfly with this equation,

$$F(t) \; = \iint_S P\boldsymbol{n}\,\mathrm{d}s \; + \iint_S \tau_w \cdot \boldsymbol{n}\,\mathrm{d}s, \tag{2.8}$$

in which $P$ is pressure, $\tau_w$ is the wall shear tensor, $\boldsymbol{n}$ denotes the normal vector of the butterfly surface, $s$ is the area of the surface. In each time step, after the aerodynamic forces were solved, we calculated the acceleration on dividing the instantaneous aerodynamic forces by mass. The flight velocity $V$ in the next step was then updated on integrating the instantaneous acceleration to unit time step ($t_2 - t_1$)

$$V(t_2) \; - V(t_1) = \int_{t_1}^{t_2} \frac{F_X}{m}\,\mathrm{d}t \; \boldsymbol{i} + \int_{t_1}^{t_2} \left(\frac{F_Y}{m} \; - g\right)\mathrm{d}t \; \boldsymbol{j}, \tag{2.9}$$

The instantaneous flight velocity of a butterfly was hence reproduced on repeating the above process in each time step in the simulation. Figure 4 compares the flight velocity between the experiment and the simulation. The wing-beat cycle of the simulation data in figure 4 was chosen after the butterfly model

attained a periodic stability, in which the aerodynamic force became stable and did not alter in further cycles.

### 2.4.2. Importing the velocity distribution

In our discussion of the aerodynamic force (§4.1), we imported the velocity distribution to ensure that all cases have the same relative flow. A relative coordinate system was adopted in which the origin moved translationally with the centre of mass of the butterfly model relative to the ground. This condition makes the model experience an inlet velocity that is opposite to the flying speed of a butterfly (§2.3). The imported inlet velocity was the simulation flight speed in figure 4 based on the real kinematic motions. Some researchers made the butterfly model fly at a constant speed or applied a fixed flow in their simulation [43,44] to make their butterfly model fly at the same velocity to compare the flow-field conditions; this method possibly neglects the effect of a transient flight speed, especially for a butterfly that has an obvious transient speed during its flight [41]. In our research, we adopted a transient flight velocity (figure 4) instead of imposing a constant flight speed. The benefit of this method is that we made all cases of the butterfly model fly with the same input velocity without neglecting the transient effect, which improves the accuracy of the simulation outcome.

## 3. Results

### 3.1. Kinematics

A butterfly was set in an observation box in a stationary condition and allowed to fly freely. The kinematics of a butterfly derived from the method in §§2.1 and 2.2 are shown in figure 5. The butterfly has an asymmetric flapping (figure 5a), in which the duration of a downstroke is longer than that of an upstroke (about 0.6–0.4 periods). In this research, the amplitude peak to peak of the wing-pitch angle in an individual cycle is determined as $\alpha_A$, which is about 45° for a real flying butterfly (figure 5d). The wing-pitch angle of the butterfly in our experiment is divisible into four stages in chronological order: slow pronation, rapid supination, slow supination and rapid pronation. In the slow pronation stage, the butterfly is at the beginning of the downstroke; the wings pitch slightly downward for about 25% of the overall amplitude of the wing-pitch angle as the wings flap downward. At $t^* = 0.3$–$0.5$, the butterfly experiences a rapid supination stage; the wings supinate for about 75% of the overall amplitude in only 0.2 period. In the slow supination stage, the wings supinate for about 25% of the overall amplitude; the supination takes 0.3 period to attain the maximum wing-pitch angle in a cycle. Thereafter, the butterfly begins pronation at the mid-upstroke; it pronates rapidly for about 75% of the overall amplitude until the end of the cycle.

The fitted curves of these kinematics approximated with expansion in a Fourier series (as $n = 3$) appear in figure 5. The equations of the fitted curves based on the real kinematic motion of a butterfly are shown as below; $t^*$ is normalized time $t/T$

$$\theta(t^*), = 29.92 - 3.66\cos(2\pi t^*) - 7.30\sin(2\pi t^*) - 1.71\cos(4\pi t^*) + 0.89\sin(4\pi t^*)$$
$$+ 0.46\cos(6\pi t^*) - 1.12\sin(6\pi t^*) \tag{3.1}$$

$$\phi(t^*) = 18.95 - 55.06\cos(2\pi t^*) + 1.76\sin(2\pi t^*) - 2.30\cos(4\pi t^*) - 7.50\sin(4\pi t^*)$$
$$+ 0.48\cos(6\pi t^*) + 0.76\sin(6\pi t^*) \tag{3.2}$$

$$\psi(t^*) = 35.90 - 10.92\cos(2\pi t^*) + 0.98\sin(2\pi t^*) + 3.07\cos(4\pi t^*) - 2.87\sin(4\pi t^*)$$
$$+ 1.07\cos(6\pi t^*) - 0.57\sin(6\pi t^*) \tag{3.3}$$

and
$$\alpha(t^*) = 10.66 - 6.62\cos(2\pi t^*) - 22.37\sin(2\pi t^*) + 0.92\cos(4\pi t^*) - 1.05\sin(4\pi t^*)$$
$$- 2.83\cos(6\pi t^*) + 0.18\sin(6\pi t^*). \tag{3.4}$$

### 3.2. Aerodynamic forces

The aerodynamic forces (figure 6a,b) were derived from the numerical model according to the real body and wing kinematics of a butterfly (equations (3.1)–(3.4)). We varied the amplitude of wing-pitch angle $\alpha_A$ and left other parameters unaltered. The number in 'wp0.25' means that $\alpha_A$ is 25% of the measured value ($\alpha_A = 45°$; figure 5d). The number measured from a real flying butterfly is hence denoted as 'wp1.00'; that set $\alpha_A = 67.5°$ is denoted as 'wp1.50'. We defined $\alpha_A$ less than 45° as 'small $\alpha_A$' and $\alpha_A$

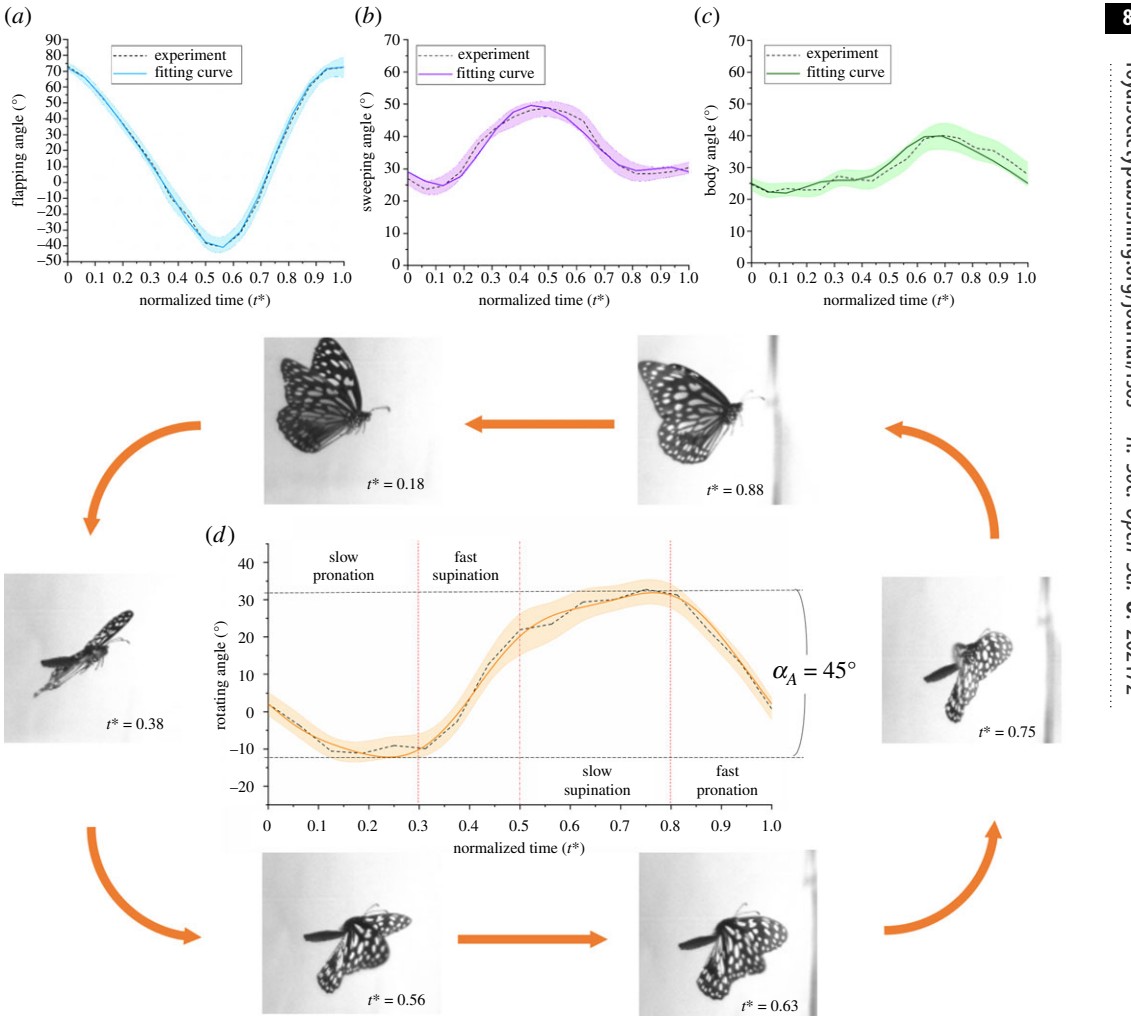

**Figure 5.** Kinematic functions of (a) flapping, (b) sweeping, (c) body and (d) wing-pitch angle. The broken lines with error bars are experimental data; the shaded areas are standard error of mean (s.e.m.); the solid lines are fitted curves. The wing-pitch angle of a butterfly comprises slow pronation, rapid supination, slow supination and rapid pronation in chronological order. The shaded area represents the downstroke, which lasts for 0.6 period.

that exceeding 45° as 'large $\alpha_A$'. To make all cases fly at the same velocity without neglecting the transient effect, we adopted a distribution of input velocity instead of imposing a constant flight speed (see §2.4).

The lift and thrust curves for these seven cases are shown in figure 6a,b. As the body-pitch angle is small in the downstroke, the wing flaps forward and downward so as to generate a positive lift and a negative thrust; during an upstroke, as the body-pitch angle is large, the wing flaps backward and upward so as to generate a negative lift and a positive thrust. This result is in accordance with previous research [5,13]. At $t^* = 0.40$ and $t^* = 0.72$, there were large differences among the cases on the force curve. The maximum magnitude of the lift (figure 6a) occurred in case wp1.50 at $t^* = 0.40$ and in case wp0.00 at $t^* = 0.72$. The maximum magnitude of thrust (figure 6b) occurred in case wp1.50 at $t^* = 0.40$; cases wp0.00 to wp1.00 showed almost the same thrust at $t^* = 0.72$. The position and angular velocity of the wing-pitch angle are shown in figure 6c,d. In figure 6c, the positions of the wing-pitch angle among the various cases have almost the same value, $t^* = 0.40$, and their positions differ at $t^* = 0.60$–$0.80$; the largest wing-pitch angle occurred in case wp1.50 at $t^* = 0.80$. In figure 6d, the wing-pitch rate has a maximum magnitude in case wp1.50 at $t^* = 0.40$; the value is almost the same at $t^* = 0.60$–$0.80$. We found that the lift at $t^* = 0.40$ and 0.72, and the thrust at $t^* = 0.40$, showed a monotonically increasing or decreasing trend with varied wing-pitch angle and wing-pitch rates, but the thrust at $t^* = 0.72$ failed to conform to a trend. This discrepancy indicates that the trend of thrust during $t^* = 0.60$–$0.80$ might arise from another mechanism; we discuss the aerodynamics occurring at this moment in §4.2.

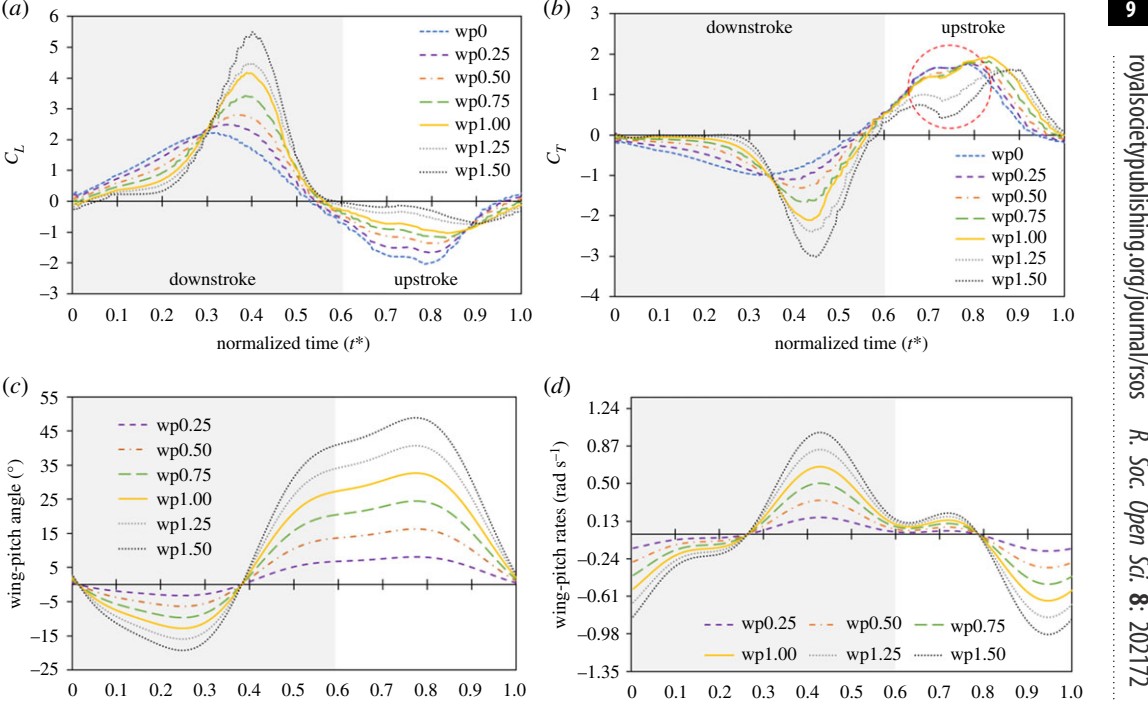

**Figure 6.** Analysis of $\alpha_A$ on transient (a) $C_L$ and (b) $C_T$, (c) wing-pitch angle and (d) wing-pitch rates (angular velocity) of a butterfly based on real kinematic motions. The large differences at $t^* = 0.72$ are marked in the red circle.

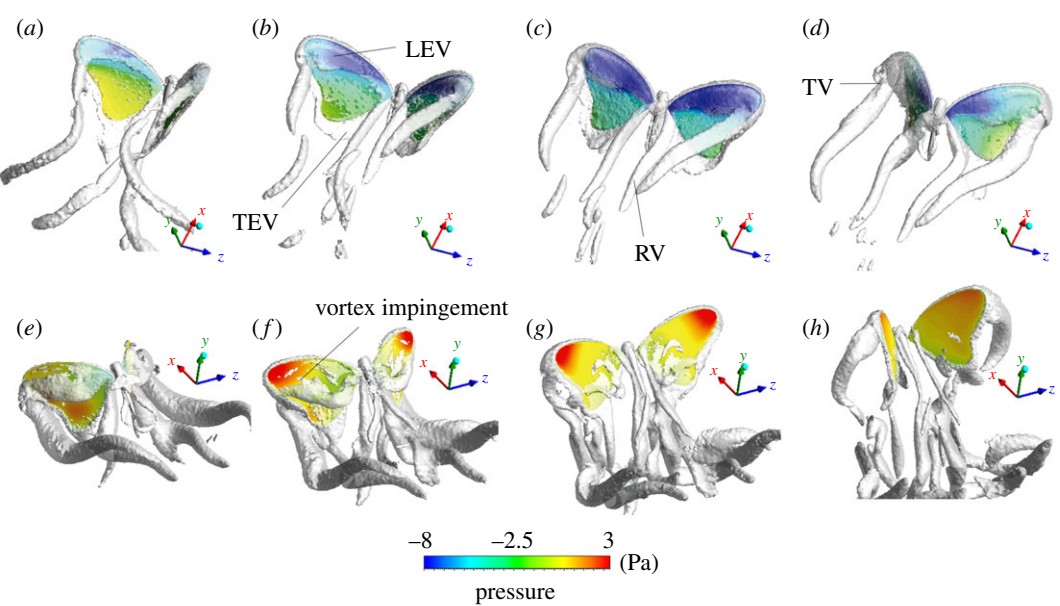

**Figure 7.** Three-dimensional flow features and contours of wing pressure of case wp1.00. The value of iso-surface $Q$ is 20000. (a–d) The downstroke flow at $t^* = 0.2$, 0.3, 0.4, 0.5; (e–h) the upstroke flow at $t^* = 0.6$, 0.7, 0.8, 0.9, respectively. TV, tip vortex; RV, root vortex.

## 3.3. Flow features

The sequences of case wp1.00 in figure 7 describe the pressure contours on the wing surface. The three-dimensional flow features around the butterfly were investigated also with the iso-surface of the $Q$-criterion. At the beginning of the downstroke, an LEV formed on the wing (figure 7a), which caused a low-pressure area on the leading edge. The region of low pressure grew during the downstroke, and attained a maximum at $t^* = 0.40$ (figure 7c). The LEV merged into the tip vortex (TV) and detached from the wing at the downstroke reversal, at which we found almost no area of low

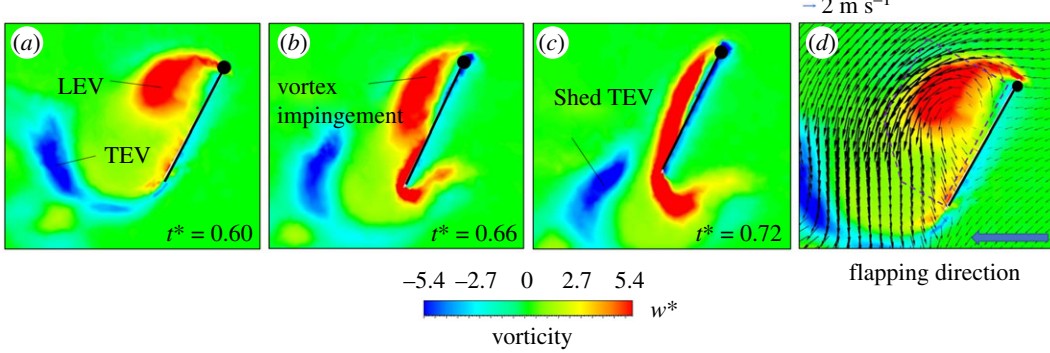

**Figure 8.** Vorticity contours from case wp1.00 with (a) $t^* = 0.60$, (b) $t^* = 0.66$ and (c) $t^* = 0.72$. (d) The figure displays the velocity vector of figure 8a.

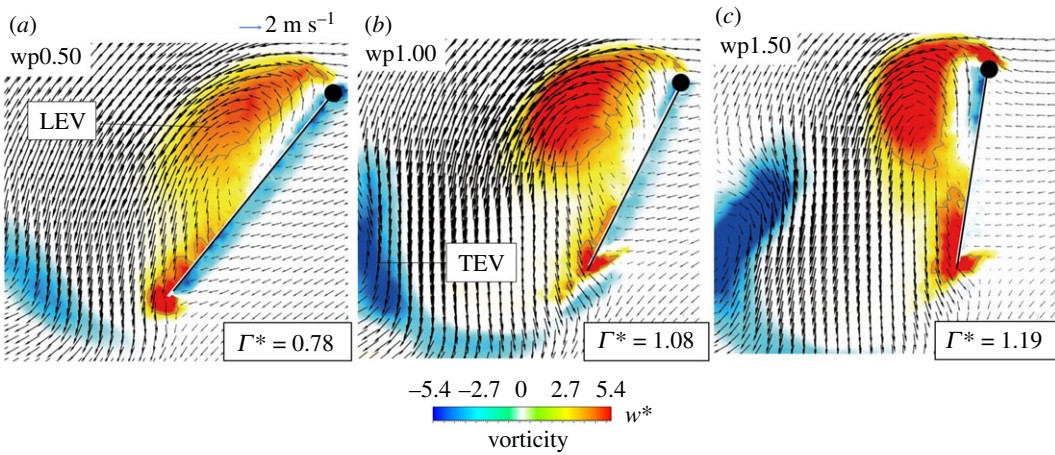

**Figure 9.** Vorticity contours and velocity vectors from cases (a) wp0.50, (b) wp1.00 and (c) wp1.50 with $t^* = 0.60$. The length of a vector represents the magnitude of the velocity. The circulation is normalized with the mean wing-tip velocity and mean wing chord ($\Gamma^* = \Gamma/U_{tip}\bar{c}$).

pressure on the wing surface (figure 7e). The wings impinged on the detached vortex at $t^* = 0.60$–0.70, and broke the structures of the shed vortex (figure 7f,g). Most high pressure was concentrated at the wing-tip during the upstroke. At the end of the stroke ($t^* = 0.90$; figure 7h), the two wings clapped at the dorsal side; the high-pressure region on the wing surface contracted.

We show contour plots of the spanwise component of vorticity around the wings of case wp1.00 at the beginning of the upstroke ($t^* = 0.60, 0.66, 0.72$). Figure 8a–c displays the spanwise vorticity contours in the chord-wise plane that coincides with the position of the radius of gyration ($r_2$), which is about 53% of the wing length. The wing is just at rest and preparing to move to the left at the downstroke reversal ($t^* = 0.60$) in figure 8a. The LEV and trailing-edge vortex (TEV) are seen around the wing surface. In figure 8b,c, the wing is moving to the left and impinges on the detached downstroke LEV. The direction of flow of the shed vortex in figure 8d illustrates the velocity vector around the wing at the beginning of the upstroke ($t^* = 0.60$) for case wp1.00. We see a detached LEV that is counter-clockwise prevailing on the wing surface; the induced airflow is generated by the LEV, and the TEV flows towards the wing surface. As the direction of the induced flow is opposite to the movement of the wing, it enhances the relative flow to the wings [11,26].

The vorticity contour, the velocity flow field and circulation $\Gamma$ in the plane of position $r_2$ in cases wp0.50, wp1.00 and wp1.50 are shown in figure 9. The circulation of the shed LEV is quantified with

$$\Gamma = \iint_s \boldsymbol{\omega} \cdot d\boldsymbol{S}, \tag{3.5}$$

in which $\boldsymbol{\omega}$ is the vorticity; the bounded area $S$ is encompassed with a vorticity contour line of value 4.6 (grey line in figure 9). A larger vorticity forms a greater circulation, which generates a larger induced flow behind the wing. The LEV and TEV formed behind the wing are weak in case wp0.50;

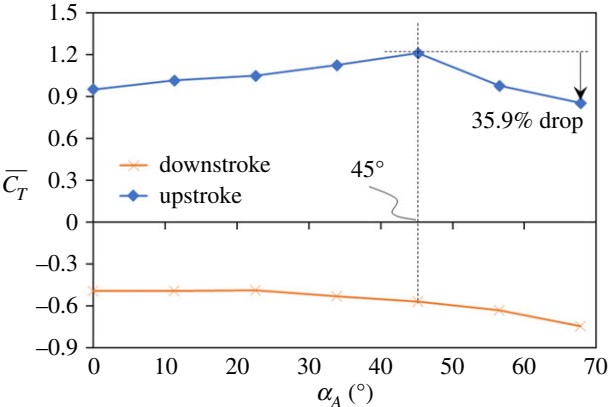

**Figure 10.** $\overline{C_T}$ during downstroke and upstroke. $\overline{C_T}$ decreases at large $\alpha_A$. Case $\alpha_A = 67.5°$ has decreased 35.9% in the upstroke relative to case $\alpha_A = 45°$.

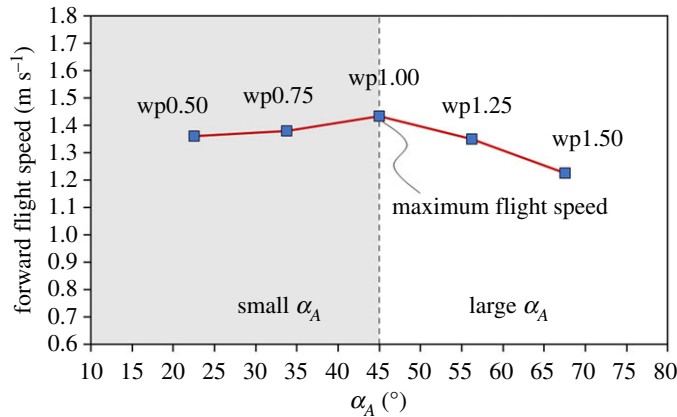

**Figure 11.** Forward flight speed with wp0.50 ($\alpha_A = 22.5°$), wp0.75 ($\alpha_A = 33.75°$), wp1.00 ($\alpha_A = 45°$), wp1.25 ($\alpha_A = 56.25°$) and wp1.50 ($\alpha_A = 67.5°$) based on real kinematic motions. Case wp1.00 flies farther than the other cases.

$\Gamma$ in case wp0.50 is also the weakest. As the pitch rate increases, the shed LEV and TEV become stronger; $\Gamma$ increases. The induced flow alters from flowing towards the wake (figure 9a,b) to a strong downwash flow vertical to the ground (figure 9c); the position of the wing also becomes more nearly vertical.

## 3.4. Average thrust force

The average thrust force ($\overline{C_T}$) in the downstroke and upstroke in each case is shown in figure 10. The horizontal axis is the amplitude of the wing-pitch angle. The average thrust force in a downstroke is marked as a cross; the upstroke is marked as a diamond. A positive thrust force is generated during an upstroke, but becomes negative during a downstroke. During a downstroke, as $\alpha_A$ increases, $\overline{C_T}$ slightly decreases. During an upstroke, as $\alpha_A$ increases, $\overline{C_T}$ increases only when $\alpha_A$ is less than 45° (wp1.00) and attains a maximum at $\alpha_A = 45°$. If $\alpha_A$ increases further, $\overline{C_T}$ begins to decrease. Compared with $\alpha_A = 45°$ (the real wing-pitch amplitude measured from experiment; figure 5d), $\overline{C_T}$ in case $\alpha_A = 67.5°$ decreases 35.9% during an upstroke.

## 3.5. Free flight

In previous sections, the velocity inlet (figure 3d) of all cases is the same in the simulation. Here, we display the flight speed of a butterfly model in a free-flight simulation based on the real kinematics (figure 5). The free-flight simulation method is described in §2.4. Five cases—wp0.50, wp0.75, wp1.00, wp1.25 and wp1.50—were applied. All five cases had already become stable, such that the flight velocity and forces did not alter in the succeeding cycle. In figure 11, the average forward flight speed during a cycle increases at small $\alpha_A$, and attains a maximum (1.43 m s$^{-1}$) at $\alpha_A = 45°$ (case wp1.00).

The average forward flight speed decreases to 1.35 and 1.23 m s$^{-1}$ at large $\alpha_A$ (case wp1.25 and wp1.50), which means that, for small $\alpha_A$, the butterfly experiences a larger forward propulsion and for large $\alpha_A$, a butterfly loses a part of the forward propulsion.

# 4. Discussion

## 4.1. Analysis of lift and thrust with varied wing-pitch motion

A wing-pitch motion is commonly observed in insects [15,17], and is also described for butterflies [6,13,22,23]. The amplitude of the wing-pitch motion ($\alpha_A$) of a butterfly, which is about 50° [13], is much smaller than that of other insects, such as a cicada, 100° [14], a dragonfly, 135° [15], a hawkmoth, 95° [16], a mosquito, 145° [17], or a fruit fly, 135° [18]. Some researchers postulated the reason to be that the fore and hind wings of a butterfly partially overlap, limiting the degree of freedom of the wings [19,20]. Other authors stated the wing-pitch angle to be unnecessary because a body-pitching oscillation plays a role equivalent to that of wing rotation [7]. It is of interest to ascertain the reason that a butterfly flies with a small $\alpha_A$, and what would occur on flying with a larger $\alpha_A$.

The lift and thrust curves for varied $\alpha_A$ are shown in figure 6a,b. The magnitudes of $C_L$ (figure 6a) and $C_T$ increase (figure 6b) as $\alpha_A$ increases at $t^* = 0.40$. As figure 6c shows, the positions of the wing-pitch angle for varied $\alpha_A$ have almost the same value at this moment; the differences at $t^* = 0.40$ arise from the wing-pitch rates. In figure 6d, the extremum of the wing-pitch rate occurs at this moment. The rapid pitch rotation of an insect is reported to enhance the aerodynamic forces with an additional rotational force [12,24,45], which was subsequently confirmed [46,47], and discussed for insects [11,17,48]. The variation at $t^* = 0.40$ is due to a rotational force generated along with a rapid pitch rotation. At $t^* = 0.60$–0.80, the magnitude of $C_L$ decreases as $\alpha_A$ increases. As seen in figure 6d, the wing-pitch rates have almost the same value during this interval; the differences at $t^* = 0.60$–0.80 arise from the position of the wing-pitch angle. In figure 6c, the position of the wing-pitch angle at $t^* = 0.60$–0.80 increases as $\alpha_A$ increases. As at large $\alpha_A$, the large value of the wing-pitch angle makes the wing align almost vertical to the ground during the upstroke, which decreases the vertical component of the aerodynamic force. The decreasing magnitude of $C_L$ as $\alpha_A$ increases at $t^* = 0.60$–0.80 is hence due to the position of the wing-pitch angle.

As the wing is more nearly vertical to the ground at large $\alpha_A$, it not only decreases the magnitude of $C_L$ but also increases $C_T$. The magnitude of $C_T$ at $t^* = 0.60$–0.80 does not, however, increase as $\alpha_A$ increases (figure 6b). The propulsion even substantially decreases at large $\alpha_A$ (cases wp1.25 and wp1.50). This effect cannot be explained with the pitch angle and pitch rate of the wing; an unconventional unsteady mechanism occurs at this instant.

## 4.2. Aerodynamic effects during the downstroke reversal stage

Most forward propulsion of a butterfly arises during the upstroke (figure 6b) [5,6], as then the wings flap backward and move in a direction opposite to that of the body when a butterfly is flying forward, which might greatly weaken the relative flow to the wings and decrease the aerodynamic force. Unsteady mechanisms occur at the downstroke reversal stage ($t^* = 0.60$) that increases the propulsion of the butterfly. During this stage, the influence of the aerodynamics is divisible into rapid pitch rotation, delayed stall of the LEV and wake capture [45]. As the rates of wing pitch (figure 6d) in each case have almost the same value at this moment, a rapid pitch rotation does not produce this result. The delayed stall of the LEV is insignificant at this moment, because the LEV generated in the downstroke proceeds to detach from the upper wing surface; the LEV generated in the upstroke has not formed at the lower surface (figure 8). The variation of the flow field is thus affected mainly by the wake-capture effect. The wake capture is that, when the wing encounters the induced flow generated by a previous downstroke or upstroke vortex, the induced flow alters the relative velocity, thus leading to increased aerodynamic forces [11,45]. We see in figure 8a–c that the wings impinge on the detached LEV generated in the downstroke. This detached vortex leads to an induced flow that is opposite to the movement of the wing (figure 8d), which enhances the relative flow to the wing and increases the forward propulsion. The wing flaps into the wake that has fluid flow towards the wing, and captures this induced flow to generate an additional aerodynamic force [25,29,49].

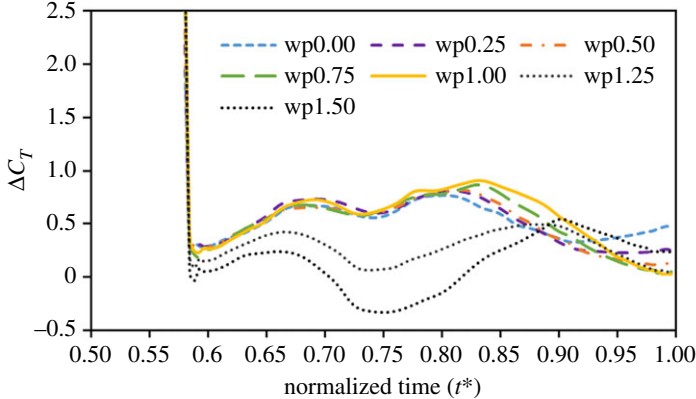

**Figure 12.** Isolated wake capture of varied $\alpha_A$. $\Delta C_T$ is the normalized $F_{wc}$.

The butterfly captures the induced flow at the downstroke reversal ($t^* = 0.60$) to gain an additional propulsion, and generates a larger thrust force during the upstroke. The force curve with varied $\alpha_A$ (figure 6b) shows however that, for large $\alpha_A$ (cases wp1.25 and wp1.50), the thrust force substantially decreases at the beginning of the upstroke (figure 6b), which leads to a smaller average thrust coefficient (figure 9).

We investigated the flow-field conditions at the stroke reversal stage ($t^* = 0.60$) around the wing in figure 9. The effect of the wake capture combines with the wing position and the flow structure around the wings [12,26,32]. In figure 9a–c, as the pitch rates increase, the direction of the induced flow behind the wing alters from diagonal rear to vertical; the position of the wing-pitch angle also varies from oblique to vertical. In figure 9a, although the direction of the induced flow is diagonal rear of the butterfly wing, the position of the wing is oblique, so that the wing captures the induced flow. In case wp1.00, despite the position of the wing being more nearly vertical than for wp0.50, the wing also captures the induced flow because the stronger circulation of the shed LEV leads the induced airflow to flow more downward. In case wp1.50, although the induced airflow flows the most downward, the wing is almost vertical to the ground. The induced flow in this case thus drifts away from the wing instead of flowing towards the wing. This analysis shows that the wake-capture effect is notable at small $\alpha_A$, but weak at large $\alpha_A$, which explains that a thrust force in cases wp0.00, wp0.25, wp0.50, wp0.75 and wp1.00 at $t^* = 0.60–0.80$ is larger than that in cases wp1.25 and wp1.50 (figure 6b).

## 4.3. Isolated wake-capture effect

Using a smoke-wire visualization, Srygley & Thomas [31] found that a butterfly uses a variation of an unsteady mechanism in preceding strokes. Although their research found a wake capture, in a particular situation, the butterfly did not capture the wake at the beginning of the upstroke. In our work, the butterfly benefits from the wake capture at small $\alpha_A$, but the effect decreases at large $\alpha_A$ as the butterfly does not capture the induced flow.

To ensure that the variation of $C_T$ (figure 6b) at the beginning of the upstroke is dominated by a wake-capture effect, we isolated the wake-capture effect on comparing the force obtained from the tenth wing-beat cycle and the force without wake capture. All seven cases were executed as follows:

$$F_{wc} = F_{tenth} - F_{initial}, \tag{4.1}$$

in which $F_{wc}$ is an additional force caused by the wake-capture effect; $F_{tenth}$ is the original force during the 10th wing-beat cycle. $F_{initial}$ is the force when the wings are initially positioned in still air at the stage of downstroke reversal with the corresponding kinematic motion, and impulsively begin flapping until the end of the half-stroke. As there is no wake behind the wing, $F_{initial}$ is uncontaminated by the wake effect [49]; $F_{wc}$ hence represents the force caused by the wake capture. In figure 12, $F_{wc}$ in small $\alpha_A$ (cases wp0.00, wp0.25, wp0.50, wp0.75 and wp1.00) is almost at the same positive value, and decreases markedly in cases wp1.25 and wp1.50. To quantify the results, we found that $F_{wc}$ in small $\alpha_A$ accounts for more than 47% of the original thrust force, $F_{tenth}$, and $F_{wc}$ decreases in proportion in large $\alpha_A$ (table 2). The butterfly hence benefits from the wake capture at small $\alpha_A$.

In the free-flight simulation (figure 11), we found that the average speed of forward flight during a cycle increases at small $\alpha_A$, and attains a maximum at $\alpha_A = 45°$ (case wp1.00), which is exactly what is derived from a real flying butterfly. The forward flight speed begins to decrease when $\alpha_A$ exceeds 45°

**Table 2.** Averaged forces of $F_{tenth}$, $F_{initial}$ and $F_{wc}$ during upstroke.

| | wp0.00 | wp0.25 | wp0.50 | wp0.75 | wp1.00 | wp1.25 | wp1.50 |
|---|---|---|---|---|---|---|---|
| $F_{tenth}$ ($10^{-3}$ N) | 2.53 | 2.71 | 2.79 | 3.00 | 3.22 | 2.61 | 2.28 |
| $F_{initial}$ ($10^{-3}$ N) | 1.12 | 1.30 | 1.45 | 1.59 | 1.70 | 1.84 | 1.92 |
| $F_{wc}$ ($10^{-3}$ N) | 1.41 | 1.41 | 1.34 | 1.41 | 1.52 | 0.77 | 0.35 |
| percentage of $F_{wc}$ | 55.7% | 52.0% | 48.1% | 47.0% | 47.1% | 29.6% | 15.6% |

(see electronic supplementary material, figure S1). Our analysis shows that a butterfly adopts a small amplitude of wing-pitch motion to enhance the wake-capture effect so as to increase the forward propulsion. When a butterfly flies with larger $\alpha_A$, the wing positions and the flow structure around the wings at the beginning of the upstroke weaken the wake-capture effect, which remarkably decreases the forward propulsion. This effect might elucidate why the amplitude of the wing-pitch angle of a butterfly is smaller than that of other insects.

## 5. Conclusion

We analysed the effect of the wing-pitch angle on the flight performance of a butterfly. Based on a numerical simulation refined with experimental data, we show that, for $\alpha_A < 45°$, the position of the wing and the detached LEV and TEV in the downstroke lead the induced airflow in the wake region to flow towards the wings, which increases the magnitude of the airflow velocity relative to the wings. At the beginning of the upstroke, the butterfly captures this induced flow and gains an additional thrust force. By contrast, for $\alpha_A > 45°$, the greater magnitude of the LEV and TEV causes the induced airflow in the wake region to drift away from the wings, which leads to a decreased relative flow to the wings, so that the thrust force decreases. These consequences are perceptible in the free-flight simulation. For small $\alpha_A$, the speed of forward flight increases, but begins to decrease for $\alpha_A > 45°$. According to the literature, a butterfly flies with a small amplitude of wing-pitch angle [6,13,22,23]; our analysis provides an essential aerodynamic feature for a butterfly to adopt a small amplitude of the wing-pitch motion to enhance the wake-capture effect so as to increase forward propulsion. The wing-pitch motion is conventionally considered to benefit the lift force [12,24,45], but a large amplitude of the motion decreases the speed of forward flight, which might be useful for the design of wing kinematics for a micro-aerial vehicle.

In this work, we parametrized the wing-pitch angle and clarified the variation of the flow field correlated with the wing-pitch motion. The other parameters of the wing and body kinematics might impose an influence on the flow field of a butterfly; this aspect remains for consideration in our future work.

Data accessibility. Data used for figures 3c, 4–6 and 10–12 in this paper are available from the Dryad Digital Repository: https://doi.org/10.5061/dryad.pzgmsbcjt [50].

Authors' contributions. Y.-J.L. conceived and designed the study, carried out the simulation and drafted the manuscript. S.-K.C. coordinated the study and helped to draft the manuscript. Y.-H.L. and J.-T.Y. contributed to the experimental design and revised the paper critically for intellectual content. J.-T.Y. is the P.I. of the projects and supervises the research progress. All authors contributed critically to writing the manuscript and gave final approval for publication.

Competing interests. We declare we have no competing interests.

Funding. National Taiwan University under contract NTU-CC-109L893301 and Taiwan Ministry of Science and Technology supported this work under a project of contract MOST 109-2221-E-002-201-MY2.

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
