## [Peer Review File · Royal Society Open Science]

Review History

RSOS-202172.R0 (Original submission)

Review form: Reviewer 1 (Ayodeji Bode-Oke)

Is the manuscript scientifically sound in its present form?

Yes

Are the interpretations and conclusions justified by the results?

Yes

Is the language acceptable?

Yes

Do you have any ethical concerns with this paper?

No

Have you any concerns about statistical analyses in this paper?

No

Recommendation?

Accept with minor revision (please list in comments)

Comments to the Author(s)

Comments attached in file (see Appendix A).

Review form: Reviewer 2

Is the manuscript scientifically sound in its present form?

Yes

Are the interpretations and conclusions justified by the results?

Yes

Is the language acceptable?

Yes

Do you have any ethical concerns with this paper?

No

Have you any concerns about statistical analyses in this paper?

No

Recommendation?

Major revision is needed (please make suggestions in comments)

Comments to the Author(s)

Please see attachment (see Appendix B).

Decision letter (RSOS-202172.R0)

Dear Dr Lin

The Editors assigned to your paper RSOS-202172 "Beneficial wake-capture effect for forward propulsion with a restrained wing-pitch motion of a butterfly" have now received comments from reviewers and would like you to revise the paper in accordance with the reviewer comments and any comments from the Editors. Please note this decision does not guarantee eventual acceptance.

Please submit your revised manuscript and required files (see below) no later than 21 days from today's (ie 14-May-2021) date. Note: the ScholarOne system will 'lock' if submission of the revision is attempted 21 or more days after the deadline. If you do not think you will be able to meet this deadline please contact the editorial office immediately.

on behalf of Dr Jake Socha (Associate Editor) and Kevin Padian (Subject Editor)
openscience@royalsociety.org

Associate Editor Comments to Author (Dr Jake Socha):

Comments to the Author:

Both reviewers had positive views of the study, but also provided a number of technical suggestions to improve the manuscript. Please pay careful attention to address their concerns in your next revision.

Reviewer comments to Author:

Reviewer: 1

Comments to the Author(s)

Comments attached in file (Review Response.pdf)

Reviewer: 2

Comments to the Author(s)

Please see attachment. (Review-RSOS-202172.pdf)

===PREPARING YOUR MANUSCRIPT===

a 'clean' version of the new manuscript that incorporates the changes made, but does not highlight them. This version will be used for typesetting if your manuscript is accepted. Please ensure that any equations included in the paper are editable text and not embedded images.

===PREPARING YOUR REVISION IN SCHOLARONE===

- If you are requesting a discretionary waiver for the article processing charge, the waiver form must be included at this step.
- If you are providing image files for potential cover images, please upload these at this step, and inform the editorial office you have done so. You must hold the copyright to any image provided.
- A copy of your point-by-point response to referees and Editors. This will expedite the preparation of your proof.

- Ensure that your data access statement meets the requirements at <https://royalsociety.org/journals/authors/author-guidelines/#data>. You should ensure that you cite the dataset in your reference list. If you have deposited data etc in the Dryad repository, please include both the 'For publication' link and 'For review' link at this stage.
- If you are requesting an article processing charge waiver, you must select the relevant waiver option (if requesting a discretionary waiver, the form should have been uploaded at Step 3 'File upload' above).
- If you have uploaded ESM files, please ensure you follow the guidance at <https://royalsociety.org/journals/authors/author-guidelines/#supplementary-material> to include a suitable title and informative caption. An example of appropriate titling and captioning may be found at https://figshare.com/articles/Table_S2_from_Is_there_a_trade-off_between_peak_performance_and_performance_breadth_across_temperatures_for_aerobic_scooping_in_teleost_fishes_/3843624.

Author's Response to Decision Letter for (RSOS-202172.R0)

See Appendix C.

RSOS-202172.R1 (Revision)

Review form: Reviewer 1 (Ayodeji Bode-Oke)

Is the manuscript scientifically sound in its present form?

Yes

Are the interpretations and conclusions justified by the results?

Yes

Is the language acceptable?

Yes

Do you have any ethical concerns with this paper?

No

Have you any concerns about statistical analyses in this paper?

No

Recommendation?

Accept as is

Comments to the Author(s)

The authors have addressed all my comments and provided acceptable justifications to the comments. I recommend the paper for publication.

Review form: Reviewer 2

Is the manuscript scientifically sound in its present form?

Yes

Are the interpretations and conclusions justified by the results?

Yes

Is the language acceptable?

Yes

Do you have any ethical concerns with this paper?

No

Have you any concerns about statistical analyses in this paper?

No

Recommendation?

Accept with minor revision (please list in comments)

Comments to the Author(s)

I appreciate the revisions made by the authors in response to my comments. While the response to my comments is nicely done, some of these need to be integrated into the manuscript for the readers. The following points need to be addressed before I can recommend publication.

1) Section 2.2: Wing tip vector has been defined in response to a comment in my previous review. The authors have also discussed the general ambiguity in defining the pitch axis from existing literature, and presented a series of images to show how they obtained the 12 deg. angle between wingtip vector and pitch axis in their study. However, none of this material is presented in the manuscript. I would like the authors to include the figure showing experimental measurement of wing pitch axis on specimens and the supporting text in the manuscript.

If the authors do not want to increase the page/figure count in the manuscript, they can include this content as supplementary material and cross-refer to that in the manuscript text in section 2.2.

2) It would be useful to add a short overall description of the simulation design at the start of section 2.4, before launching into specific details in subsections 2.4.1 and 2.4.2. In this description, outline the reasons for separating the modelling efforts into two parts (i.e., free-flight simulation based on experimentally prescribed body pitch and wing kinematics, importing velocity distribution).

Also, please describe the outputs from each step (i.e., what was calculated from each step) in the manuscript. I am still not clear as to whether forces were estimated from the free-flight simulations or the prescribed flight speed simulations. It would be useful to indicate which figure/result was obtained from which type of simulation in the Results section.

3) In subsection 2.4.2, please add details of the relative coordinate system and origin being translated based on the COM of the butterfly relative to the ground.

4) The discussion of the reasoning for negative lift and negative thrust in Figure 6 needs to be included in the manuscript text under section 3.2.

Decision letter (RSOS-202172.R1)

Dear Dr Lin

On behalf of the Editors, we are pleased to inform you that your Manuscript RSOS-202172.R1 "Beneficial wake-capture effect for forward propulsion with a restrained wing-pitch motion of a butterfly" has been accepted for publication in Royal Society Open Science subject to minor revision in accordance with the referees' reports. Please find the referees' comments along with any feedback from the Editors below my signature.

Please submit your revised manuscript and required files (see below) no later than 7 days from today's (ie 21-Jul-2021) date. Note: the ScholarOne system will 'lock' if submission of the revision is attempted 7 or more days after the deadline. If you do not think you will be able to meet this deadline please contact the editorial office immediately.

on behalf of Dr Jake Socha (Associate Editor) and Kevin Padian (Subject Editor)
openscience@royalsociety.org

Associate Editor Comments to Author (Dr Jake Socha):

Comments to the Author:

Congratulations on acceptance of this manuscript! The revisions largely satisfied the reviewers. Prior to publication, we recommend that you address the additional items brought up by reviewer 2, because they should help to strength the final paper.

Reviewer comments to Author:

Reviewer: 1

Comments to the Author(s)

The authors have addressed all my comments and provided acceptable justifications to the comments. I recommend the paper for publication.

Reviewer: 2

Comments to the Author(s)

I appreciate the revisions made by the authors in response to my comments. While the response to my comments is nicely done, some of these need to be integrated into the manuscript for the readers. The following points need to be addressed before I can recommend publication.

1) Section 2.2: Wing tip vector has been defined in response to a comment in my previous review. The authors have also discussed the general ambiguity in defining the pitch axis from existing literature, and presented a series of images to show how they obtained the 12 deg. angle between wingtip vector and pitch axis in their study. However, none of this material is presented in the manuscript. I would like the authors to include the figure showing experimental measurement of wing pitch axis on specimens and the supporting text in the manuscript.

If the authors do not want to increase the page/figure count in the manuscript, they can include this content as supplementary material and cross-refer to that in the manuscript text in section 2.2.

2) It would be useful to add a short overall description of the simulation design at the start of section 2.4, before launching into specific details in subsections 2.4.1 and 2.4.2. In this description, outline the reasons for separating the modelling efforts into two parts (i.e., free-flight simulation based on experimentally prescribed body pitch and wing kinematics, importing velocity distribution).

Also, please describe the outputs from each step (i.e., what was calculated from each step) in the manuscript. I am still not clear as to whether forces were estimated from the free-flight simulations or the prescribed flight speed simulations. It would be useful to indicate which figure/result was obtained from which type of simulation in the Results section.

3) In subsection 2.4.2, please add details of the relative coordinate system and origin being translated based on the COM of the butterfly relative to the ground.

4) The discussion of the reasoning for negative lift and negative thrust in Figure 6 needs to be included in the manuscript text under section 3.2.

===PREPARING YOUR MANUSCRIPT===

===PREPARING YOUR REVISION IN SCHOLARONE===

- An editable file of each table (.doc, .docx, .xls, .xlsx, or .csv).
- An editable file of all figure and table captions.

- Any electronic supplementary material (ESM).
- If you are requesting a discretionary waiver for the article processing charge, the waiver form must be included at this step.
- If you are providing image files for potential cover images, please upload these at this step, and inform the editorial office you have done so. You must hold the copyright to any image provided.
- A copy of your point-by-point response to referees and Editors. This will expedite the preparation of your proof.

- Ensure that your data access statement meets the requirements at <https://royalsociety.org/journals/authors/author-guidelines/#data>. You should ensure that you cite the dataset in your reference list. If you have deposited data etc in the Dryad repository, please only include the 'For publication' link at this stage. You should remove the 'For review' link.
- If you are requesting an article processing charge waiver, you must select the relevant waiver option (if requesting a discretionary waiver, the form should have been uploaded at Step 3 'File upload' above).
- If you have uploaded ESM files, please ensure you follow the guidance at <https://royalsociety.org/journals/authors/author-guidelines/#supplementary-material> to include a suitable title and informative caption. An example of appropriate titling and captioning may be found at https://figshare.com/articles/Table_S2_from_Is_there_a_trade-off_between_peak_performance_and_performance_breadth_across_temperatures_for_aerobic_scope_in_teleost_fishes_/3843624.

Author's Response to Decision Letter for (RSOS-202172.R1)

See Appendix D.

Decision letter (RSOS-202172.R2)

Dear Dr Lin,

I am pleased to inform you that your manuscript entitled "Beneficial wake-capture effect for forward propulsion with a restrained wing-pitch motion of a butterfly" is now accepted for publication in Royal Society Open Science.

on behalf of Dr Jake Socha (Associate Editor) and Kevin Padian (Subject Editor)
openscience@royalsociety.org

Appendix A

Review Response: Beneficial wake-capture effect for forward propulsion with a restrained wing-pitch motion of a butterfly

Compared to other insects, the wing pitch motion of a butterfly is of a lesser amplitude. Thus, this article focused on investigating how the small pitch amplitude affects the aerodynamics of the butterfly in forward flight. The authors performed a parametric study (based on numerical simulations based on both free flight and prescribed flight (with velocity distribution) based on high-speed videos), systematically increasing the pitch angle. The authors showed that if high amplitude pitch motion were possible, wake capture benefits may be attenuated and forward propulsion will be reduced. The study is an interesting one and well-structured and provides insights into aerodynamics of butterfly forward flight with reduced pitch amplitudes.

With the consideration and incorporation of the comments below, the manuscript can be further strengthened.

Here are some comments that may be useful for the authors in the final manuscript.

Minor Comments

1. **Page -2, Lines 36 -37.** MAV). “A butterfly with a small wing-beat frequency thus becomes an ideal model for the design of a MAV”. I am not quite convinced that the butterfly is an ideal model for an MAV solely based on flapping frequency, since flapping frequency is not the only consideration in MAV design. Furthermore, other advanced insect-inspired MAV’s were not inspired by a butterfly, e.g., The Robobee and Delfly. I’d suggest revising or removing this statement.
2. **Page 2, Line 51-54.** “The amplitude of the wing-pitch angle of a butterfly is, however, much smaller than that of other insects, ...”. Please include the numerical value of the amplitudes, for easier comparison. Previous studies by Zheng et al. [1] and Bode-Oke et al. [2], contain information to extract the wing pitch amplitude of a butterfly in flight.
3. **Page 2, Line 53- Page 3 line 3-6.** The authors list studies where wing pitch motion was ignored in the analysis, but as in the above comment, there are studies [1,2] that did not neglect the wing pitch motion in the analysis and are worth mentioning to give a more accurate description of studies that have been conducted, hitherto. The uniqueness of the current work is that the wing pitch amplitude is systematically varied via a parametric study as opposed to [1,2] that simply had the pitch angles obtained from the butterfly in their study (i.e. they only evaluated $w_p=1.0$).
4. **Page 4, line 13-16.** “...The characteristic points of wing-tip and mid-hindwing were chosen inside the wing surface rather than at the outer edge to mitigate the influence of wing flexibility on the measurement of wing-pitch angle...”. This suggests that the wing pitch was calculated at only one cross section. Is that so? If so, why wasn’t a reference point such as the radius of gyration used as the reference location for calculating wing pitch? Furthermore, wing pitch changes along the span, especially in the upstroke due to upstroke twist (see Fig 5 in [2]). Thus, will the wake-capture finding be different if another cross-section (which will have a different amplitude from 45 degs) was used to get the wing pitch that was then used for the flat wing model?

5. **Page 6. Section 2.3.** How long does a typical simulation take, given the computational resources used in this study?

Major Comments

Wake capture is an auxiliary lift enhancing mechanism when compared to Delayed/Absence of stall which is responsible for most of the force generation in insect flight. Thus, it is less likely that pitch amplitude will be sacrificed just for the sake of wake capture. At small pitch amplitudes, this study found, that the butterfly performs better (thrust-wise) and takes advantage of wake capture but this is only in a low-speed range (max forward speed is 1.4m/s, fig 4), we do not know what happens at higher speeds. Nevertheless, lift at the lower pitching amplitudes were smaller than at large pitching amplitudes, too (fig 6). Srygley and Thomas [3], showed that butterflies use varied aerodynamic mechanisms in flight. *V. atalanta*, did not use wake capture in every wing beat of a forward flight sequence in their study, thus, it may not be as important as suggested in providing “a physical elucidation for the fact that a butterfly adopts a small amplitude of the wing-pitch motion for flight”. Hence, I do not think wake capture alone can confirm why a small pitch amplitude is adopted, as other constraints other than aerodynamics play a huge role too.

References

1. Zheng, L., Hedrick, T. L., & Mittal, R. (2013). Time-varying wing-twist improves aerodynamic efficiency of forward flight in butterflies. *PloS one*, 8(1), e53060.
2. Bode-Oke, A. T., & Dong, H. (2020). The reverse flight of a monarch butterfly (*Danaus plexippus*) is characterized by a weight-supporting upstroke and postural changes. *Journal of the Royal Society Interface*, 17(167), 20200268.
3. Srygley, R. B., & Thomas, A. L. R. (2002). Unconventional lift-generating mechanisms in free-flying butterflies. *Nature*, 420(6916), 660-664.

Appendix B

The manuscript submitted by Lin et al. describes a numerical investigation of the effects of wake-capture during forward propulsion of a butterfly. The results suggest that the small amplitude of the wing-pitch motion used by a butterfly can increase the airspeed relative to the wings via wake-capture, augmenting thrust force to enhance forward propulsion. The question examined in this study is interesting and the manuscript is generally well-written. However, some key details need to be addressed before I can recommend publication.

Major comments:

- 1) Section 2.2: Definition of wing angles – Wing tip vector definition is not clear. Why is the wing tip vector at an angle of 12° with wing pitch axis? Does wing pitch axis change between individual butterflies?
- 2) Section 2.4- P7, L16 – “.....a butterfly is let fly freely as loaded...” – Does this mean the butterfly model is freely flying and did you use moving mesh problem? If not, do the flight speeds represent “pseudo velocities” that are calculated from the forces generated on a fixed butterfly? This needs to be explained clearly as it is confusing right now.
- 3) What is the significance of negative lift and negative thrust in Figure 6? In Figure 6a, for $w_p0.00$, the average lift coefficient is close to zero. Does this mean that butterfly will generate no lift at zero pitch angle? Is this reasonable? Please add a brief discussion on this point.
- 4) Figure 6b: Although the magnitude of thrust (C_T) decreases at higher wing pitch angles, we generally see higher magnitude of thrust in the downstroke. Also, it appears that the magnitude of cycle-averaged thrust (across upstroke and downstroke) will be higher for higher pitch angle? Why would it be less preferable to use higher pitch angles in this case?
- 5) Figure 7 pressure contour scale bar is missing.
- 6) Figure 8 shows vorticity contours for upstroke motion of a wing at different time points. When the wing is performing upstroke (moving to the left in the figure), wouldn't the LEV be rotating in clockwise sense, i.e., negative vorticity in LEV and positive vorticity in TEV? Is this because of the inlet velocity used in the CFD simulations? What is the inlet velocity used in this study?
- 7) The conclusion for this study is based on the finding that induced flow drifts away from the wing at higher wing pitch angles thereby dropping wing-wake capture and therefore the aerodynamic thrust during upstroke. The relation between induced flow and wing-wake capture is not yet clear and needs to be further investigated and presented. From Figure 9, it looks like at higher wing pitch angles, the TEV wake is coming closer to the wing compared to that of lower pitch angles. Does this not make wing-wake capture significant?
- 8) The words “wing-wake capture” and “induced flow” are used interchangeably throughout the study which is very confusing while reading the manuscript. Does wake capture represents capturing the induced flow? Please try to address this issue.

Minor comments:

Abstract, P1, L20: “additional forward propulsion” is vague, not sure how this is quantified. Also, the MAV comment is a bit out of context at the very end.

P1, L35: better to use “durability” instead of “lifetime”

P1, L40: “extraordinary” and “obvious” wording sounds exaggerated, I would advise the authors to stick to specific details that make butterfly flight unique (beyond low flapping frequency)

P4, L3: “designated screen” should be shown in figure 1

P4, L9: 16 videos from 1 individual? Please clarify

P4 L13-L14: refer to both 1b and 1c

Please indicate sweep and pitch angles in figure 2

P6, L40-L41: "origin coincided with....centre of mass of the butterfly"--was the centre of mass displacement prescribed from experimental data?

Fig. 3 caption: 6S instead of 3S?

P12, L52: multiple time points during the upstroke are shown but the authors only refer to 0.60.

P14, L10: provide justification for the vorticity contour cutoff that was used

P15, L9: "for butterflies"

Was body pitching/oscillation considered in the study? Can the interaction of body pitching and wing pitching alter the study findings?

P16, L51: "leading-edge vortex and wake capture"?

Appendix C

Overview of the revision (Ms. Ref. No.: RSOS-202172)

Thank you for your attention to our revised manuscript entitled, “Beneficial wake-capture effect for forward propulsion with a restrained wing-pitch motion of a butterfly” (RSOS-202172). We respond to all comments of the Reviewers point by point, and make our manuscript clearer and readable to general readers.

According to Reviewer #1, we modified the introduction and strengthened the importance of the wake capture by quantifying the numerical value of wake capture. We also analyzed the possible transient aerodynamic effects individually, so as to exclude their influence in the response.

According to Reviewer #2, we have revised the errors and modified the abstract that the Reviewer considered vague in the revised manuscript. The definition of the wake capture is added accordingly in the revised manuscript. We also adequately respond to the technical questions.

Overall Response to the comments of Reviewer #1

We are grateful for the valuable and constructive suggestions, which improve the article. We have accordingly modified the part of introduction, and respond to the comments point by point.

The Reviewer questions that the wake capture might not be the only reason for a butterfly to use a small amplitude of wing-pitch angle. It is true that a real flying butterfly flies in a complicated manner; there could be a factor other than wake capture that contributes to the small amplitude of wing-pitch angle. We find, however, that the wake capture is a crucial mechanism for a butterfly to generate thrust. In a small wing-pitch amplitude, the wake capture accounts for more than 47 % of the thrust generated during an upstroke. To verify the importance of wake capture, we further analyzed the possible unsteady flapping mechanisms (rapid pitch rotation, delayed stall of the leading-edge vortex, the tip vortex effect and wake capture) individually at the stage of stroke reversal ($t^* = 0.60$), and quantified the influences of each mechanism; the results at greater flight speed were also investigated. We believe that the wake capture is one essential aerodynamic feature for a butterfly to adopt a small amplitude of the wing-pitch motion.

We really appreciate the valuable suggestion. All changes are marked in red in the revised manuscript. The detailed responses have also been made point by point in the specific response list.

Specific response to the comments of Reviewer #1

Minor Comments

1. Page -2, Lines 36 -37. MAV). “A butterfly with a small wing-beat frequency thus becomes an ideal model for the design of a MAV”. I am not quite convinced that the butterfly is an ideal model for an MAV solely based on flapping frequency, since flapping frequency is not the only consideration in MAV design. Furthermore, other advanced insect-inspired MAV’s were not inspired by a butterfly, e.g., The Robobee and Delfly. I’d suggest revising or removing this statement.

Response:

Thank you for the comment. We have changed the word “an ideal model” to “an effective biomimetic model” in the revised manuscript.

2. Page 2, Line 51-54. “The amplitude of the wing-pitch angle of a butterfly is, however, much smaller than that of other insects, ...”. Please include the numerical value of the amplitudes, for easier comparison. Previous studies by Zheng et al. [1] and Bode-Oke et al. [2], contain information to extract the wing pitch amplitude of a butterfly in flight.

Response:

We have added a citation about the numerical value of the wing-pitch amplitude of butterflies for comparison on lines 41 and 387 in the revised manuscript.

3. Page 2, Line 53- Page 3 line 3-6. The authors list studies where wing pitch motion was ignored in the analysis, but as in the above comment, there are studies [1,2] that did not neglect the wing pitch motion in the analysis and are worth mentioning to give a more accurate description of studies that have been conducted, hitherto. The uniqueness of the current work is that the wing pitch amplitude is systematically varied via a parametric study as opposed to [1,2] that simply had the pitch angles obtained from the butterfly in their study (i.e. they only evaluated $w_p=1.0$).

Response:

We have added a citation that did not neglect the wing-pitch motion, and added some description in lines 47-50 in the revised manuscript.

4. Page 4, line 13-16. “...The characteristic points of wing-tip and mid-hindwing were chosen inside the wing surface rather than at the outer edge to mitigate the influence of wing flexibility on the measurement of wing-pitch angle...”. This suggests that the wing pitch was calculated at only one cross section. Is that so? If so, why wasn’t a reference point such as the radius of gyration used as the reference location for calculating wing pitch? Furthermore, wing pitch changes along the span, especially in the upstroke due to upstroke twist (see Fig 5 in [2]). Thus, will the wake-capture finding

be different if another cross-section (which will have a different amplitude from 45 degs) was used to get the wing pitch that was then used for the flat wing model?

Response:

This suggestion to choose the radius of gyration as a reference point is appreciated. A different cross section mainly affects the amplitude of the wing-pitch angle (figure 1c in Bomphrey *et al.*, 2017; figure 5a in Bode-Oke & Dong, 2020). There is no difference in our final results if we choose another cross section that leads to a different amplitude, as what we are doing is a parametric study. This parametric study in our research systematically analyzes amplitude from 0° to 67.5°, which includes the entire range of wing-pitch amplitude of a butterfly derived from varied wing cross section. The trend of the aerodynamic forces hence remains unchanged in our simulation if an amplitude different from 45° were used. To strengthen our idea, we chose amplitude 83.25° that is beyond the range 0° to 67.5° in Fig. 1. The trend of the aerodynamic forces of the amplitude at 83.25° is in accordance with other cases.

Fig. 1. Aerodynamic thrust forces with variation of amplitude of wing pitch ranging from 0° to 83.25°.

5. Page 6. Section 2.3. How long does a typical simulation take, given the computational resources used in this study?

Response:

The simulation for a typical case takes about 15 h to calculate a period.

Major Comments

Wake capture is an auxiliary lift enhancing mechanism when compared to Delayed/Absence of stall which is responsible for most of the force generation in insect flight. Thus, it is less likely that pitch amplitude will be sacrificed just for the sake of wake capture. At small pitch amplitudes, this study

found, that the butterfly performs better (thrust-wise) and takes advantage of wake capture but this is only in a low-speed range (max forward speed is 1.4m/s, fig 4), we do not know what happens at higher speeds. Nevertheless, lift at the lower pitching amplitudes were smaller than at large pitching amplitudes, too (fig 6). Srgyley and Thomas [3], showed that butterflies use varied aerodynamic mechanisms in flight. *V. atalanta*, did not use wake capture in every wing beat of a forward flight sequence in their study, thus, it may not be as important as suggested in providing “a physical elucidation for the fact that a butterfly adopts a small amplitude of the wing-pitch motion for flight”. Hence, I do not think wake capture alone can confirm why a small pitch amplitude is adopted, as other constraints other than aerodynamics play a huge role too.

Response:

In our work, we investigated the relation between wing-pitch angle and the flow field generated by a butterfly. We found that an atypical trend of thrust force occurred during $t^* = 0.60-0.80$ if a varied wing-pitch amplitude was adopted. This phenomenon has not been clarified in previous articles. We thus focused on the impact of the fluid flow on the butterfly with varied wing-pitch amplitude during an upstroke, and discovered that the wake capture at stroke reversal ($t^* = 0.60$) is the main factor leading to the variation of thrust. Although the delayed stall is an important mechanism to generate lift for insects, the wake capture is also crucial to generate thrust for a butterfly. Most forward propulsion is produced in an upstroke (figure 10 in the revised manuscript); for a small wing-pitch amplitude, the wake capture accounts for more than 47 % of the generated thrust during an upstroke (figure 12 and Table 2 in revised manuscript). The wake capture is a critical mechanism for a butterfly to generate thrust. To verify the importance of the wake capture, we analyzed the possible unsteady flapping mechanisms (i.e. rapid pitch rotation, delayed stall of the leading-edge vortex, the tip vortex effect and wake capture) individually at the stage of the stroke reversal ($t^* = 0.60$), and quantified the influences of each mechanism.

1. Rapid pitch rotation

The rapid pitch-up rotation of the wing leads to an increased vorticity around the wing (Shyy *et al.*, 2008), and generates an augmented lift. To quantify the mechanism of the rapid pitch rotation, we calculated the strength of the circulation around the wing,

$$\Gamma = \iint_S \vec{\omega} \cdot d\vec{S}$$

in which $\vec{\omega}$ is the vorticity. Three normalized times ($t^*=0.60, 0.66$ and 0.72) were chosen; the chord-wise section that coincides with the position of the radius of gyration (r_2) was selected. The bounded area of the surface integral is defined as a circle of

radius 0.025 m and centred in the middle of the wing section (Fig. 2a).

According to previous research, the rotational force has a positive relation with the wing-pitch angular velocity (Sane & Dickinson, 2002), and is also proportional to the circulation in the surrounding air (Chin & Lentink, 2016). In our case, we found that, in table 1, the value of the circulation varies little among cases wp0.50, wp1.00 and wp1.50 at $t^* = 0.60, 0.66$ and 0.72 , which indicates that the wing is unaffected by the mechanism of the rapid pitch rotation at the stroke reversal. This result is reasonable as the angular velocity of the wing-pitch angle has almost the same value during that moment ($t^* = 0.60-0.72$) in our work (Fig. 2b). The outcomes indicate that the rotational circulation can be neglected at the stage of stroke reversal.

Fig. 2. (a) Bounded area of the surface integral. (b) Angular velocity of varied amplitude of the wing-pitch motion

Table 1. Circulation around the wings in cases wp0.50, wp1.00 and wp1.50 at $t^*=0.60, 0.66$ and 0.72 .

Γ (m^2s^{-1})	wp0.50	wp1.00	wp1.50
$t^* = 0.60$	0.1124	0.1187	0.1179
$t^* = 0.66$	0.1185	0.1248	0.1229
$t^* = 0.72$	0.1265	0.1232	0.1262

2. Delayed stall of leading-edge vortex

The performance of the delayed stall of a leading-edge vortex (LEV) is affected seriously by the attachment of the LEV (Shyy & Liu, 2007). At the stage of the stroke reversal ($t^* = 0.60$), at which the wake capture occurs, the LEV generated in the downstroke proceeds to detach from the wing; the LEV generated in the upstroke has not formed at the lower surface (Fig. 3a). Liang *et al.* (2010) stated that the LEV was not obvious at the beginning of the upstroke. From the pressure distribution (Fig. 3b),

there is no area of low pressure on the lower surface, which indicates that the LEV in the upstroke has not formed at the lower surface. On the upper surface, although a slight low pressure exists near the wing root, this pressure is weak (-5 Pa); there is no low pressure near the wing tip. This condition indicates that the influence of the attachment of the LEV was insignificant at this moment. We hence considered that the delayed stall of the leading-edge vortex is not the main factor to affect the aerodynamic force at the stroke reversal.

Fig. 3. (a) Vorticity contour around the wing and (b) pressure distribution on the wing surface of cases wp0.50, wp1.00 and wp1.50 at the stroke reversal ($t^*=0.60$).

3. Tip vortex

For a three-dimensional (3D) flapping wing of small aspect ratio, the vortex generates an area of low pressure near the wing tip and interacts with the LEV to influence the aerodynamic production of force (Shyy *et al.*, 2009). The effect of tip vortex is excluded in the case of two-dimensional (2D) flow (Sane, 2003); we hence undertook a 2D simulation with the same flying conditions to investigate whether the variation of the wake capture at the beginning of the upstroke ($t^* = 0.6-0.72$) exists with the effect of the tip vortex completely excluded.

The 2D wing chord is chosen as the local wing chord located at the position of the radius of gyration (r^2). The fluid domain was set as 20 times the wing length. The amplitude of the wing-pitch angle remained the same as in our 3D case. The results appear in Fig. 4.

Fig. 4. Analysis of variation of amplitude of the wing-pitch angle on the transient thrust force in 2D and 3D simulations

At $t^* = 0.60-0.72$, the thrust curve in a 2D simulation also decreases at cases wp1.25 and wp1.50 compared with a 3D simulation (figure 4), which represents that, without the existence of the tip vortex, the thrust also varies with the varying amplitude of the wing-pitch angle. The effect of the tip vortex contained in 3D appears to affect not seriously the outcomes of the 2D simulation.

The vorticity contours and the flow field of small α_A (case wp0.50) and large α_A (case wp1.50) at stroke reversal ($t^* = 0.60$) are shown in Fig. 5. As the amplitude of the wing-pitch angle increased, the induced flow generated by the LEV and TEV (trailing-edge vortex) altered from flowing towards the wing surface to a strong downwash flowing away from the wing, leading to a separate level of the wake-capture effect, the same as the results in the 3D simulation. On excluding the effect of the tip vortex, the variation of the flow field correlated with the wing-pitch motion remained unchanged. We thus consider that the tip vortex does not affect the results in our work.

Fig. 5. Velocity flow field around the wing for small (wp0.50) and large (wp1.50) amplitudes of wing-pitch motions

For a butterfly, Zheng *et al.* (2013) measured a flying speed 1.14 m s^{-1} , Yokoyama *et al.* (2013), 1.6 m s^{-1} , and Kang *et al.* (2018), 1.77 m s^{-1} . Most butterflies fly in the range 1 m s^{-1} to 2 m s^{-1} . To investigate the effect of wing-pitch angle at increased flight speed, we executed the simulation with the maximum flight speed, 2.0 m s^{-1} ; the transient aerodynamic thrust forces are shown in Fig. 6a. We see that the force values at maximum flight speed 2.0 m s^{-1} (Fig. 6a) are less than at maximum flight speed 1.4 m s^{-1} (Fig. 6b). The reason is that the kinematic motion loaded to the butterfly model is measured from the real experiment that has maximum flight speed 1.4 m s^{-1} ; maximum flight speed 2.0 m s^{-1} is too great for the model. Even at a greater flight speed, however, the aerodynamic thrust force is larger in a small wing-pitch amplitude during an upstroke; the thrust force decreases at increased amplitude of the wing-pitch angle. The results are consistent with smaller flight speeds, which indicates that at greater flight speeds, the thrust force also increases with a small wing-pitch amplitude.

Fig. 6. Transient aerodynamic thrust forces with maximum flight speed (a) 2.0 m s^{-1} , (b) 1.4 m s^{-1} .

Regarding the research of Srygley & Thomas (2002), they used a smoke-wire visualization to investigate the fluid flow of a real freely flying butterfly. They reported that a butterfly did not use wake capture in the sequence of flight. According to our research, we consider that this condition occurred in their study was because of the varied wing-pitch motion during the experiment. The above examination of the transient mechanism analysis and the aerodynamic forces at higher speed shows that, when adopting a small wing-pitch amplitude, the wake capture is a main mechanism for a butterfly to increase the thrust force. We hence believe the wake capture is an essential aerodynamic mechanism to generate thrust for a butterfly. Despite that condition, we also agree that there might be other factors other than aerodynamics contributed to the small wing-pitch angle. We thus changed the description of "a physical elucidation" to "one essential aerodynamic features" in the revised manuscript in line 18, 81 and 496.

References

- A. T. Bode-Oke and H. Dong, "The reverse flight of a monarch butterfly (*Danaus plexippus*) is characterized by a weight-supporting upstroke and postural changes," *J. R. Soc. Interface* 17, 20200268 (2020).
- C. K. Kang, J. Cranford, M. K. Sridhar, D. Kodali, D. B. Landrum, and N. Slegers, "Experimental characterization of a butterfly in climbing flight," *AIAA J.* 56, 15-24 (2018).
- D. D. Chin and D. Lentink, "Flapping wing aerodynamics: from insects to vertebrates," *J. Exp. Biol.* 219, 920-932 (2016).

- L. Zheng, T. L. Hedrick, and R. Mittal, "Time-varying wing-twist improves aerodynamic efficiency of forward flight in butterflies," *PLoS One* 8, e53060 (2013).
- N. Yokoyama, K. Senda, M. Iima, and N. Hirai, "Aerodynamic forces and vortical structures in flapping butterfly's forward flight," *Phys. Fluids* 25, 021902 (2013).
- R. B. Srygley and A. L. R. Thomas, "Unconventional lift-generating mechanisms in free-flying butterflies," *Nature* 420, 660-664 (2002).
- R. J. Bomphrey, T. Nakata, N. Phillips, and S. M. Walker, "Smart wing rotation and trailing-edge vortices enable high frequency mosquito flight," *Nature* 544, 92-95 (2017).
- S. P. Sane, "The aerodynamics of insect flight," *J. Exp. Biol.* 206, 4191-4208 (2003).
- S. P. Sane and M. H. Dickinson, "The aerodynamic effects of wing rotation and a revised quasi-steady model of flapping flight," *J. Exp. Biol.* 205, 1087-1096 (2002).
- W. Shyy and H. Liu, "Flapping wings and aerodynamics lift: the role of leading- edge vortices," *AIAA J.* 45, 2817-9 (2007).
- W. Shyy, P. Trizila, C. K. Kang, and H. Aono, "Can tip vortices enhance lift of a flapping wing?" *AIAA J.* 47, 289-293 (2009).
- W. Shyy, Y. Lian, J. Tang, D. Vieru, and H. Liu, *Aerodynamics of low Reynolds number flyers* (Cambridge University Press, New York USA, 2008).
- Z. Liang, H Dong, M. Wei "Computational analysis of hovering hummingbird flight," In: 48th AIAA aerospace sciences meeting including the new horizons forum and aerospace exposition, (AIAA, Orlando, FL USA, 2010).
-

Overall response to the comments of Reviewer #2

We are grateful for the valuable and constructive suggestions, which improve the article. According to the comments, we corrected some errors and improved the manuscript to be clearer and readable.

In the research method, we simulated a freely flying butterfly, of which the lift and thrust were calculated with the simulator. The wing and body-pitch kinematic motion in the simulation is given based on the experimental measurement. The flight speed is calculated by the lift and thrust. To save the calculation time, we adopted a relative coordinate system in which the origin moved translationally with the centre of mass of the butterfly model. This condition makes the model experience an inlet velocity that is opposite the flying speed of a butterfly.

In our work, we varied the amplitude of the wing-pitch angle based on the real motion and left the other parameters unaltered with the same inlet velocity, to investigate the effect of wing-pitch angle in the simulation. The result shows that the butterfly benefits from the wake capture at a small amplitude of the wing-pitch angle. The wake capture is defined as, when the wing encounters the induced flow generated by a previous downstroke or upstroke vortex, the induced flow alters the relative velocity, thus leading to increased aerodynamic forces. Our definition is equivalent to that in previous articles (Sane, 2003; Shyy *et al.*, 2008; Shyy *et al.*, 2010; Lua *et al.*, 2011; Lee & Lua, 2018).

We really appreciate the valuable suggestion. All changes are marked in red in the revised manuscript. The detailed responses are made point by point in the specific response list.

Specific response to the comments of Reviewer #2

Major comments:

1) Section 2.2: Definition of wing angles – Wing tip vector definition is not clear. Why is the wing tip vector at an angle of 12° with wing pitch axis? Does wing pitch axis change between individual butterflies?

Response:

Thank you for the question. We have added the definition of the wing-tip vector in line 132 in the revised manuscript, which is set as the vector between the wing-tip and wing-root points on the wing.

For the wing-pitch axis of an insects, it is difficult to confirm where the axis is located. As the wing has a three-dimensional motion and typically moves in a complicated

way, the wing-pitch axis is different even within the same species. Bomphrey et al. (2017) reported that the wing-pitch axis moves from leading to trailing edge during flight of a mosquito. Some authors simplified the wing-pitch axis and define it at the leading edge (Wu et al., 2014; Zou et al., 2019; Lai et al., 2020), whereas others defined it at one quarter (Sun & Lan, 2004; Zheng et al., 2016) or 24% (Wang & Sun, 2005) of the wing chord. For the studies that considered the wing-pitch motion of a butterfly (Zheng et al., 2013; Bode-Oke & Dong, 2020), the definition of a wing-pitch axis is unclear. To unify the calculation of the motion analysis, in this work we defined the wing-pitch axis as the vector between the third speckle (Fig. 1) and the wing root. According to the measurement, the defined wing-pitch axis and wing-tip vector has an angle near 12° ; we thus defined the wing-pitch axis is at 12° from the wing-tip vector.

Fig. 1. Experimental measurement of the wing-pitch axis on specimens

2) Section 2.4- P7, L16 – “.....a butterfly is let fly freely as loaded...” – Does this mean the butterfly model is freely flying and did you use moving mesh problem? If not, do the flight speeds represent “pseudo velocities” that are calculated from the forces generated on a fixed butterfly? This needs to be explained clearly as it is confusing right now.

Response:

Yes, it is a moving-mesh problem. The motion in the simulation model is obtained from the experimental analysis; the mesh becomes updated according to the kinematic motion in each time step. We added this information in line 216 in the revised manuscript.

In this work, our simulation method is divided into two parts. One is free-flight simulation; the other is importing the velocity distribution.

For the first part, we simulated a free flying butterfly, in which the lift and thrust were calculated with the simulator. This method produced the result of section 3.5 in the revised manuscript. The kinematic motion (body-pitch and wing motion) in a simulation was given based on the experimental measurement. The flight speed was then integrated by the lift and thrust which was produced by the body-pitch and the wing motions of the model. Hence, the flight speed was not a pseudo-velocity issue.

For the second part, we imported the velocity distribution to make all cases of the butterfly model fly with the same input velocity. This method produced the results from

section 3.2 to 3.4 in the revised manuscript. The velocity distribution was the flight speed of butterfly, which was obtained from the flight speed of the free-flight simulation based on real wing and body-pitch kinematic motions.

As a butterfly model flies relative to the ground, a large fluid domain is required for flight and would lead to lots of time in calculations. To save the calculation time, we hence adopted a relative coordinate system in which the origin moved translationally with the centre of mass of the butterfly model relative to the ground. This condition makes the model experience an inlet velocity that is opposite the flying speed of a butterfly.

We have added subtitles 2.4.1 and 2.4.2 in the revised manuscript to make the simulation method clearer.

3) What is the significance of negative lift and negative thrust in Figure 6? In Figure 6a, for wp0.00, the average lift coefficient is close to zero. Does this mean that butterfly will generate no lift at zero pitch angle? Is this reasonable? Please add a brief discussion on this point.

Response:

In our work, we measured the real kinematic motion (body-pitch and wing motion) of a butterfly and loaded it to the simulator to calculate the lift and thrust forces. As the body-pitch angle is small in the downstroke, the wing flaps forward and downward so as to generate a positive lift and a negative thrust; during an upstroke, as the body-pitch angle is large, the wing flaps backward and upward so as to generate a negative lift and a positive thrust. There is thus a negative lift in the downstroke and a negative lift in the upstroke. This result is in accordance with other research (Fei & Yang, 2016; Bode-Oke & Dong, 2020).

In this work, to investigate the effect of wing-pitch angle in the simulation we varied the amplitude of the wing-pitch angle based on the real motion and left other parameters unaltered with the same inlet velocity. The forces in case wp0.00 were thus different from those of a real butterfly. That the average lift coefficient was near zero in case wp0.00 represents that, when the butterfly did not conduct a wing-pitch motion, it could not generate sufficient lift force.

4) Figure 6b: Although the magnitude of thrust (C_T) decreases at higher wing pitch angles, we generally see higher magnitude of thrust in the downstroke. Also, it appears that the magnitude of cycle-averaged thrust (across upstroke and downstroke) will be higher for higher pitch angle? Why would it be less preferable to use higher pitch angles in this case?

Response:

In the upstroke, the thrust decreased at a large wing-pitch amplitude. In figure 10 (revised manuscript), there is also a larger magnitude of the negative thrust in the downstroke at a large wing-pitch amplitude. The cycle-averaged thrust hence becomes smaller at a larger wing-pitch angle, which is not preferable for a butterfly to fly forward. Thank you for the question.

5) Figure 7 pressure contour scale bar is missing.

Response:

We appreciate your reminder. We have added the pressure contour scale bar.

6) Figure 8 shows vorticity contours for upstroke motion of a wing at different time points. When the wing is performing upstroke (moving to the left in the figure), wouldn't the LEV be rotating in clockwise sense, i.e., negative vorticity in LEV and positive vorticity in TEV? Is this because of the inlet velocity used in the CFD simulations? What is the inlet velocity used in this study?

Response:

The LEV and TEV shown in figure 8 (revised manuscript) are the vortices generated in the downstroke, so the LEV is rotating counter-clockwise. About the vortices in the upstroke, as we have the inlet velocity, the intensity of the vortices is significantly smaller than that in the downstroke. The three-dimensional flow features in figure 7 (revised manuscript) can also show that the intensity of the vortices is greater in the downstroke. This phenomenon has been reported by Liang *et al.* (2010).

The inlet velocity is the negative of the flight speed of butterfly. This flight speed was the imported velocity distribution which was calculated from the free-flight simulation based on real kinematic motions (Response #2). The benefit of this method is that we made all cases of the butterfly model fly with the same input velocity without neglecting the effect of transient flight speed, which is important for a butterfly (Fei & Yang, 2015).

7) The conclusion for this study is based on the finding that induced flow drifts away from the wing at higher wing pitch angles thereby dropping wing-wake capture and therefore the aerodynamic thrust during upstroke. The relation between induced flow and wing-wake capture is not yet clear and needs to be further investigated and presented. From Figure 9, it looks like at higher wing pitch angles, the TEV wake is coming closer to the wing compared to that of lower pitch angles. Does this not make wing-wake capture significant?

Response:

Thank you for the question. According to Sane (2003), the wake capture is that the wing encounters an induced jet formed by the previously shed vortex, thus producing a larger aerodynamic force. Shyy *et al.* (2010) also stated that the wake capture exists such that, when the wings flap into the wake, the flow induced by a shed vortex impinges on the wings. The definition of wake capture has been added in lines 424-426 in the revised manuscript.

Indeed, an increased wing-pitch angle makes the trailing-edge vortex (TEV) approach the wing, and produces a larger induced flow by two counter-rotating vortices (LEV and TEV). The position of the wing-pitch angle is almost vertical to the ground, which makes the wing unable to capture the induced flow, so that a wake capture did not occur, thus decreasing the aerodynamic forces.

8) The words “wing-wake capture” and “induced flow” are used interchangeably throughout the study which is very confusing while reading the manuscript. Does wake capture represents capturing the induced flow? Please try to address this issue.

Response:

Yes, the wake capture means that, when the wing encounters an induced flow generated by a previous downstroke or upstroke vortex, the induced flow alters the relative velocity, thus increasing the aerodynamic forces. Our definition is the same as in previous articles (Sane, 2003; Shyy *et al.*, 2008; Shyy *et al.*, 2010; Lua *et al.*, 2011; Lee & Lua, 2018).

Minor comments:

Abstract, P1, L20: "additional forward propulsion" is vague, not sure how this is quantified. Also, the MAV comment is a bit out of context at the very end.

Response:

Thank you for the comment. We have quantified the numerical value of the wake capture in section 4.3, and added it in the abstract to make the sentence clear. The application is also revised to specify to the design of wing kinematics of a MAV.

P1, L35: better to use "durability" instead of "lifetime"

Response:

We appreciate your suggestion. We have changed the word.

P1, L40: "extraordinary" and "obvious" wording sounds exaggerated, I would advise the authors to

stick to specific details that make butterfly flight unique (beyond low flapping frequency)

Response:

Thank you for the comment. We have modified the sentence to make it preferable.

P4, L3: "designated screen" should be shown in figure 1

Response:

Thank you for the comment. We have added it.

P4, L9: 16 videos from 1 individual? Please clarify

Response:

Sixteen videos arose from four butterflies. This information is added in line 90 in the revised manuscript.

P4 L13-L14: refer to both 1b and 1c

Response:

Thank you for the comment. We have modified it.

Please indicate sweep and pitch angles in figure 2

Response:

Thank you for the comment. The sweep and pitch angles have been added.

P6, L40-L41: "origin coincided with....centre of mass of the butterfly"--was the centre of mass displacement prescribed from experimental data?

Response:

The model in our work follows the real size of a butterfly with Solidworks. As the real density of a butterfly body is difficult to measure, we assumed that the butterfly is homogeneous in our study (Su *et al.*, 2012; Lai *et al.*, 2020; Chang *et al.*, 2021). The centre of mass is calculated from the shape of a butterfly.

Fig. 3 caption: 6S instead of 3S?

Response:

We appreciate your reminder. The radius is 3S and diameter 6S. We have modified the error.

P12, L52: multiple time points during the upstroke are shown but the authors only refer to 0.60.?

Response:

We have added other time points. Thank you for your reminder.

P14, L10: provide justification for the vorticity contour cutoff that was used

Response:

Figure 2 shows the normalized circulation for vorticity cutoff values 3.8, 4.2, 4.6 and 5.0 in cases wp0.50, wp1.00 and wp1.50. We see that the normalized circulation of all cases has the same order (wp1.50 > wp1.00 > wp0.50). This condition indicates that the choice of vorticity cutoff affects little the result; we hence chose value 4.6 as a reference. The same method was used in other research (Meng *et al.*, 2020).

Vorticity cutoff value	wp0.50	wp1.00	wp1.50
3.8			Γ^*	1.34	1.85	2.09
4.2			

Fig. 2. Normalized circulation for vorticity cutoff values 3.8, 4.2, 4.6 and 5.0 in cases wp0.50, wp1.00 and wp1.50

P15, L9: "for butterflies"

Response:

We have modified it. Thank you for the suggestion.

Was body pitching/oscillation considered in the study? Can the interaction of body pitching and wing pitching alter the study findings?

Response:

Yes, the body oscillation has been considered; its amplitude remained unchanged in this research. Our work focused mainly on the wing-pitch motion in relation to aerodynamic forces. The effect of the body angle of a butterfly has been reported in our previous articles (Fei & Yang, 2016). The interaction between body and wing-pitch angle will be our next study.

P16, L51: "leading-edge vortex and wake capture"?

Response:

Thank you for the suggestion. We have modified it.

References

- A. T. Bode-Oke and H. Dong, "The reverse flight of a monarch butterfly (*Danaus plexippus*) is characterized by a weight-supporting upstroke and postural changes," *J. R. Soc. Interface* 17, 20200268 (2020).
- D. Wu, K. S. Yeo, and T. T. Lim, "A numerical study on the free hovering flight of a model insect at low Reynolds number," *Comput. Fluids* 103, 234-261 (2014).
- J. K. Wang and M. Sun, "A computational study of the aerodynamics and forewing-hindwing interaction of a model dragonfly in forward flight," *J. Exp. Biol.* 208, 3785-3804 (2005).
- J. Y. Su, S. C. Ting, Y. H. Chang, and J. T. Yang, "A passerine spreads its tail to facilitate a rapid recovery of its body posture during hovering," *J. R. Soc. Interface* 9, 1674-1684 (2012).
- K. B. Lua, T. T. Lim, and K. S. Yeo, "Effect of wing-wake interaction on aerodynamic force generation on a 2D flapping wing," *Exp. Fluids* 51, 177-195 (2011).
- L. Zheng, T. L. Hedrick, and R. Mittal, "Time-varying wing-twist improves aerodynamic efficiency of forward flight in butterflies," *PLoS One* 8, e53060 (2013).
- M. Sun and S. L. Lan, "A computational study of the aerodynamic forces and power requirements of dragonfly (*Aeschna juncea*) hovering," *J. Exp. Biol.* 207, 1887-1901 (2004).
- P. Y. Zou, Y. H. Lai, & J. T. Yang, "Effects of phase lag on the hovering flight of damselfly and dragonfly," *Phys. Rev. E* 100, 063102 (2019).
- R. J. Bomphrey, T. Nakata, N. Phillips, and S. M. Walker, "Smart wing rotation and trailing-edge vortices enable high frequency mosquito flight," *Nature* 544, 92-95 (2017).
- S. P. Sane, "The aerodynamics of insect flight," *J. Exp. Biol.* 206, 4191-4208 (2003).
- W. Shyy, H. Aono, S. K. Chimakurthi, P. Trizila, C. K. Kang, C. E. Cesnik, and H. Liu, "Recent progress in flapping wing aerodynamics and aeroelasticity," *Prog. Aerosp. Sci.* 46, 284-327 (2010).
- W. Shyy, Y. Lian, J. Tang, D. Viieru, and H. Liu, *Aerodynamics of low Reynolds number flyers* (Cambridge University Press, New York USA, 2008).

- X. Meng, Y. Zhang, and G. Chen, "Ceiling effects on the aerodynamics of a flapping wing with advance ratio," *Phys. Fluids* 32, 021904 (2020).
- Y. H. J. Fei and J. T. Yang, "Enhanced thrust and speed revealed in the forward flight of a butterfly with transient body translation," *Phys. Rev. E* 92, 033004 (2015).
- Y. H. J. Fei and J. T. Yang, "Importance of body rotation during the flight of a butterfly," *Phys. Rev. E* 93, 033124 (2016).
- Y. H. Lai, Y. J. Lin, S. K. Chang, and J. T. Yang, "Effect of wing-wing interaction coupled with morphology and kinematic features of damselflies," *Bioinspir. Biomim.* 16, 016017 (2020).
- Y. J. Lee and K. B. Lua, "Wing-wake interaction: comparison of 2D and 3D flapping wings in hover flight," *Bioinspir. Biomim.* 13, 066003 (2018).
- Z. Liang, H Dong, M. Wei "Computational analysis of hovering hummingbird flight," In: 48th AIAA aerospace sciences meeting including the new horizons forum and aerospace exposition, (AIAA, Orlando, FL USA, 2010).

Appendix D

Overview of the revision (Ms. Ref. No.: RSOS-202172)

Thank you for your attention to our revised manuscript entitled, “Beneficial wake-capture effect for forward propulsion with a restrained wing-pitch motion of a butterfly” (RSOS-202172). We are grateful for the valuable and constructive suggestions from Reviewer #1 and Reviewer #2. We respond to all the suggestions of Reviewer #2 point by point, and revised the article to make our manuscript clearer and readable to general readers.

Specific Response to the comments of Reviewer #2

1) Section 2.2: Wing tip vector has been defined in response to a comment in my previous review. The authors have also discussed the general ambiguity in defining the pitch axis from existing literature, and presented a series of images to show how they obtained the 12 deg. angle between wingtip vector and pitch axis in their study. However, none of this material is presented in the manuscript. I would like the authors to include the figure showing experimental measurement of wing pitch axis on specimens and the supporting text in the manuscript.

If the authors do not want to increase the page/figure count in the manuscript, they can include this content as supplementary material and cross-refer to that in the manuscript text in section 2.2.

Response:

We have added the method of how to get the wing-pitch axis in the supplemental material, and added the information in manuscript text in line 132.

2) It would be useful to add a short overall description of the simulation design at the start of section 2.4, before launching into specific details in subsections 2.4.1 and 2.4.2. In this description, outline the reasons for separating the modelling efforts into two parts (i.e., free-flight simulation based on experimentally prescribed body pitch and wing kinematics, importing velocity distribution).

Also, please describe the outputs from each step (i.e., what was calculated from each step) in the manuscript. I am still not clear as to whether forces were estimated from the free-flight simulations or the prescribed flight speed simulations. It would be useful to indicate which figure/result was obtained from which type of simulation in the Results section.

Response:

Thank you for the suggestion. We have added an overview of the simulation design at the start of section 2.4 in the revised manuscript.

3) In subsection 2.4.2, please add details of the relative coordinate system and origin being translated based on the COM of the butterfly relative to the ground.

Response:

We have added this detail in line 246-248 in the revised manuscript.

4) The discussion of the reasoning for negative lift and negative thrust in Figure 6 needs to be included in the manuscript text under section 3.2.

Response:

We have added the reasoning for negative lift and thrust under section 3.2 in the revised manuscript.